# Tumour-derived PGD2 and NKp30-B7H6 engagement drives an immunosuppressive ILC2-MDSC axis

Sara Trabanelli *et al.*[#]

Group 2 innate lymphoid cells (ILC2s) are involved in human diseases, such as allergy, atopic dermatitis and nasal polyposis, but their function in human cancer remains unclear. Here we show that, in acute promyelocytic leukaemia (APL), ILC2s are increased and hyper-activated through the interaction of CRTH2 and NKp30 with elevated tumour-derived PGD2 and B7H6, respectively. ILC2s, in turn, activate monocytic myeloid-derived suppressor cells (M-MDSCs) via IL-13 secretion. Upon treating APL with all-trans retinoic acid and achieving complete remission, the levels of PGD2, NKp30, ILC2s, IL-13 and M-MDSCs are restored. Similarly, disruption of this tumour immunosuppressive axis by specifically blocking PGD2, IL-13 and NKp30 partially restores ILC2 and M-MDSC levels and results in increased survival. Thus, using APL as a model, we uncover a tolerogenic pathway that may represent a relevant immunosuppressive, therapeutic targetable, mechanism operating in various human tumour types, as supported by our observations in prostate cancer.

#A full list of authors and their affliations appears at the end of the paper.

Innate lymphoid cells (ILCs) are a family of lymphocytes involved in the initiation, regulation and resolution phases of inflammatory processes[1, 2]. ILCs differentiate into ILC1, ILC2 and ILC3 with distinct transcriptional regulation and functional attributes, which mirror the T-helper 1 (Th1), Th2 and Th17/Th22 CD4[+] lymphocytes, respectively. However, unlike T cells, ILCs lack somatically rearranged antigen receptors and lineage markers (Lin[−])[3, 4].

Originally described in murine models, ILC2 are best defined by the constitutive expression of the interleukin (IL)-7 receptor alpha chain (CD127) and the prostaglandin D2 (PGD2) receptor, CRTH2[5]. ILC2 differentiation is dependent on the transcription factors GATA3[6] and RORα[7]. Once activated by alarmins (e.g., IL-33, IL-25 and thymic stromal lymphopoietin (TSLP)) ILC2s rapidly produce effector cytokines, mostly IL-5, IL-9 and IL-13[8]. Moreover, in vitro treatment with PGD2 has been shown to induce the chemotaxis of and IL-13 production by ILC2s[9], whereas type I interferons (IFN) (mainly IFN-β), IFN-γ, IL-27[10, 11] and prostaglandin I2 (PGI2) restrain ILC2's function and suppress type 2 immunity[12]. Beside soluble mediators, ILC2s also rely on cell-cell contacts for their activation. In that context, expression of the type I Ig-like transmembrane natural cytotoxicity receptor (NCR) NKp30 on human ILC2s was shown to trigger the secretion of type 2 cytokines upon in vitro binding to one of its ligands, B7H6[13].

Dysregulation or chronic activation of ILC2s has been reported in pathologic conditions, such as allergy, atopic dermatitis and nasal polyposis[14]. However, ILC2s' function in tumour immune regulation remains largely unknown. Studies in mouse models show that ILC2s are associated with reduction in metastases in a lung metastatic tumour model, through the regulation of eosinophil recruitment[15]. In addition, ILC2 were shown to induce tumour cell apoptosis in response to locally secreted IL-33[16]. By contrast, the IL-33/IL-33 receptor (ST2) axis inhibits tumour surveillance in a breast carcinoma model by interacting with myeloid-derived suppressor cells (MDSC)[17], and promotes cholangiocyte proliferation and epithelial hyperplasia in a cholangiocarcinoma model[18]. However, the ILC2 contribution, if any, to human tumour immune responses remains unknown, with only one report showing elevated frequencies of circulating ILC2s (defined as Lin[−]ICOS[+]IL17RB[+] cells) in gastric cancer patients[19].

Among acute myeloid leukaemia (AML), acute promyelocytic leukaemia (APL) is a distinct clinico-pathologic entity characterized by the t(15;17) translocation that leads to an arrest of myeloid differentiation at the promyelocytic stage. The majority of APL patients achieve remission upon treatment by all-trans retinoic acid (ATRA) that causes the differentiation of the leukaemic clone to a post-mitotic state[20].

Here we show that ILC2s are the major ILC subtype present in human APL. Given the unique setting of a malignancy definitively cured by targeted therapies, we use APL as a model to investigate the involvement of ILC2 in human tumour establishment and clearance. We unravel a tumour immunosuppressive axis initiated by APL blasts. Via the release of PGD2 and the expression of B7H6, APL blasts engage CRTH2[+]NKp30[+] ILC2s and induce their activation and IL-13 release, which in turn drives the expansion and the immune suppressive function of IL-13Rα1[+] monocytic myeloid-derived suppressor cells (M-MDSCs). Disruption of this tumour immunosuppressive axis by specifically blocking PGD2, IL-13 and NKp30 partially normalizes ILC2 and M-MDSC levels and results in increased survival in leukaemic mice. Our additional results in prostate cancer suggest that the same axis may be activated also in solid tumours. As the identified pathways can be druggable, this axis

may have a therapeutic value in different human solid and haematologic malignancies, beyond APL.

## Results

**ILC2s are significantly increased in human APL.** Here we measured the relative and absolute numbers of ILCs in peripheral blood of 22 APL patients at diagnosis. Whereas total ILCs were comparable between healthy donors and APL patients (Supplementary Fig. 1a and Fig. 1a–c), the latter were characterized by a robust ILC1 and ILC2 enrichment in the periphery (Fig. 1d–f). However, we and others previously reported in other AML subtypes a profound ILC dysregulation affecting the ILC1 and ILC3 subsets[21, 22], with an expansion in ILC1. Since ILC1 is identified in human as ILCs negative for CRTH2, cKit and CD56 expression, we tested whether, in APL patients, the ILC1 fraction comprised also undifferentiated ILC precursors, recently defined by others as CD5[+]CD1a[−] ILCs. Within the ILC1 subset, we observed elevated levels of ILC precursors in the bone marrow of patients compared to healthy donors (Supplementary Fig. 1b, c). On the contrary, ILC2 enrichment specifically distinguishes APL from the other AML subtypes that do not show alterations in the ILC2 compartment[21, 22]. Therefore, APL represents a unique setting to delineate the cellular and molecular bases of ILC2 expansion in human tumours.

Interestingly, we found no difference in the ILC2 distribution in the bone marrow (BM) between APL patients and healthy donors (Fig. 1g, h). Still, the analysis of paired BM and peripheral blood (PB) APL samples revealed a selective increase in ILC2s in the periphery (Fig. 1i, j). To understand whether the accumulation of ILC2s in the periphery was due to increased proliferation, extended survival or both, we measured the expression of Ki-67 and Bcl-2 in these cells. Peripheral ILC2s of APL patients showed similar levels of Bcl-2 (Supplementary Fig. 1d) and increased levels of Ki-67 when compared with either peripheral ILC2s of healthy donors or BM ILC2s of APL patients (Fig. 1k).

These findings suggest that in APL patients the increased levels of circulating ILC2s are mainly due to peripheral expansion.

**PGD2 and NKp30-B7H6 binding drive IL-13 secretion by ILC2s.** ILCs exert their main functions through effector cytokines and, among them, ILC2s are mainly involved in the production of type 2 cytokines, including IL-13. Consistent with an increased frequency of peripheral ILC2s, serum concentrations of circulating IL-13 were significantly higher in APL patients than healthy donors, while the serum concentrations of circulating IFN-γ and IL-17A were comparable (Fig. 2a). Supporting an important role for ILC2s as the source of circulating IL-13, the levels of ILC2-derived IL-13 in response to co-culture with APL cell lines were significantly higher in ILCs from APL patients than controls, while no difference was observed for IFN-γ and IL-17A (Fig. 2b, c). Since IL-13 is also secreted by CD4[+] Th2 and NKT cells, we assessed by the expression of prototypic surface markers the T-helper subset distribution in APL patients as well as the presence of NKT cells. We found no difference in the levels of CD4[+] Th2 cells and NKT cells in patients compared to healthy controls (Supplementary Fig. 2a, b). When incubated with autologous primary blasts, ILC2 from APL patients, and not NKT nor Th2 cells from either group, produced higher amounts of IL-13 than ILC2s from healthy donors, and were the population with the highest capacity of producing IL-13 in response to blasts (Fig. 2d, e).

Next, we asked why circulating ILC2s produce higher amounts of IL-13 in response to APL blasts. Since IL-13 production can be triggered through the engagement of NKp30 on ILC2s by its ligand B7H6 on target cells[13], we assessed the expression of B7H6

and NKp30 in human APL blasts and ILC2s, respectively. Both the established APL cell line NB4, as well as the HL60 line, expressed high levels of the NKp30 ligand B7H6 (Fig. 2f, g). Furthermore, to ascertain whether the NKp30-B7H6 interaction could also occur in vivo, we screened primary APL blasts for the expression of B7H6. Peripheral APL blasts expressed distinctly higher levels of B7H6 than their counterparts in the

BM (Fig. 2h, i). Importantly, peripheral blood ILC2s from APL patients expressed higher levels of NKp30 when compared to matched BM counterparts as well as to ILC2s from healthy individuals (Fig. 2j, k). Consistent with these observations, in vitro blocking of the NKp30-B7H6 interaction significantly inhibited IL-13 production by ILC2s from APL patients in co-cultures with APL cells (Fig. 2l, m).

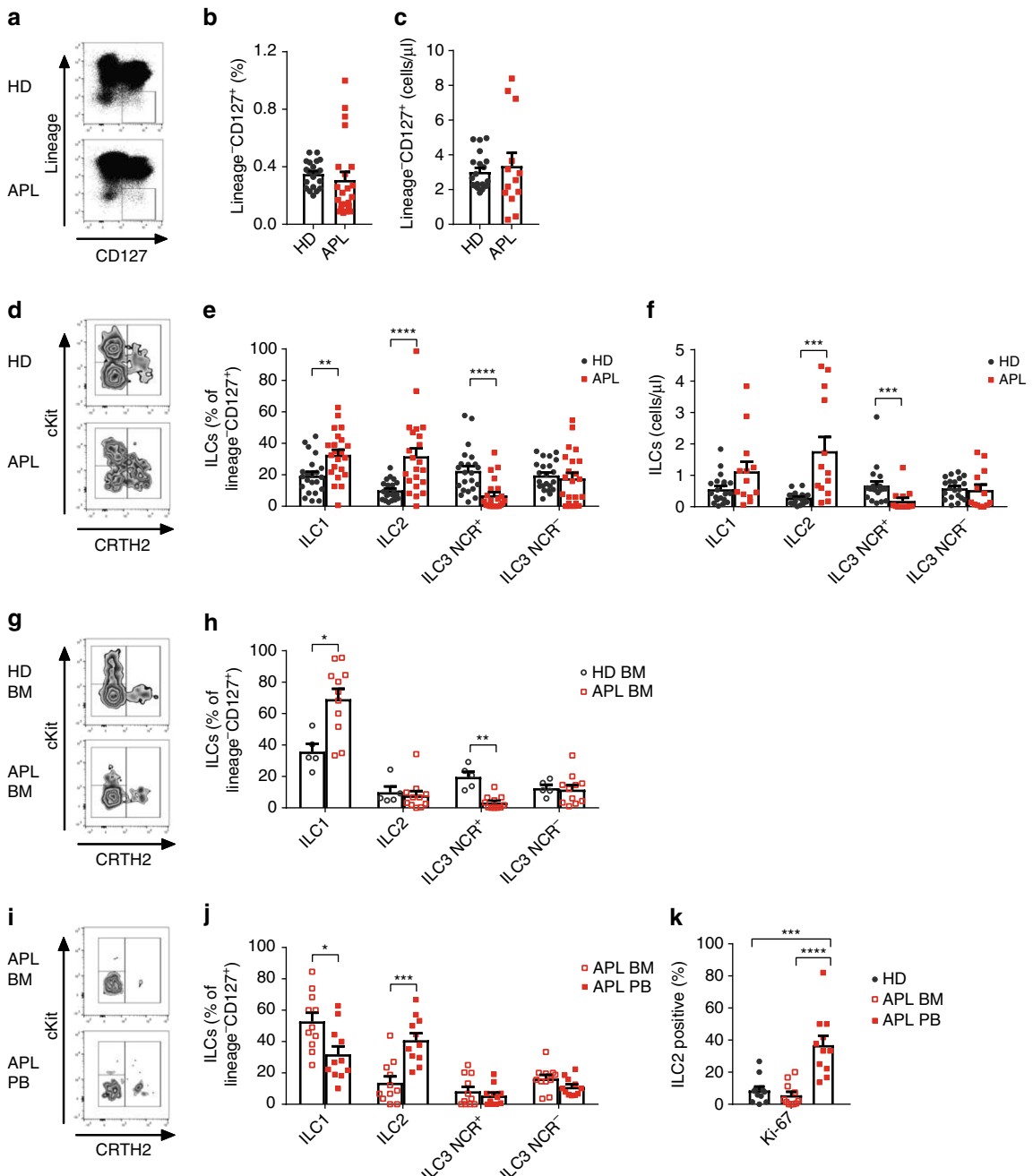

**Fig. 1** ILC2 are significantly increased in human APL. **a** Representative examples of flow cytometry analysis of innate lymphoid cells (ILC) in healthy donors (HD) and APL patients' (APL) peripheral blood. HD were sex- and age-matched with APL patients, M:F 12:10 vs. 12:10, median age 45 vs. 43, min 27 vs. 23, max 72 vs. 77 years. **b** Frequencies ($n = 22$) and **c** cell numbers/µl (HD $n = 20$, APL $n = 13$) of total ILCs identified as Lin⁻CD127⁺ cells in lymphocytes. **d** Representative examples of flow cytometry analysis of innate lymphoid cell subsets in HD and APL patients' (APL) peripheral blood. **e** Relative frequencies ($n = 22$) **f** cell numbers/µl (HD $n = 20$, APL $n = 13$) of ILC subsets among total ILCs. **g** Representative examples of flow cytometry analysis of ILC subsets in the bone marrow of healthy donors (HD BM) and APL patients (APL BM). **h** Relative frequencies of ILC subsets among total ILCs (HD $n = 5$, APL $n = 11$). **i** Representative examples of flow cytometry analysis of ILC subsets in paired APL BM and peripheral blood (APL PB) samples of APL patients. **j** Relative frequencies of ILC subsets among total ILCs ($n = 11$). **k** Frequency of Ki-67 expressing ILC2 in HD, APL BM and APL PB ($n = 11$). ILC subsets are defined as follow: ILC1: Lin⁻CD127⁺CRTH2⁻cKit⁻; ILC2: Lin⁻CD127⁺CRTH2⁺cKit⁻/⁺; ILC3 NCR⁺: Lin⁻CD127⁺CRTH2⁻cKit⁺NKp46⁺; ILC3 NCR⁻: Lin⁻CD127⁺ CRTH2⁻cKit⁺NKp46⁻. *Error bars* are s.e.m. Statistical analysis was performed using *t* test (**a–f**), Mann-Whitney test (**h–j**) and ANOVA test (**k**)

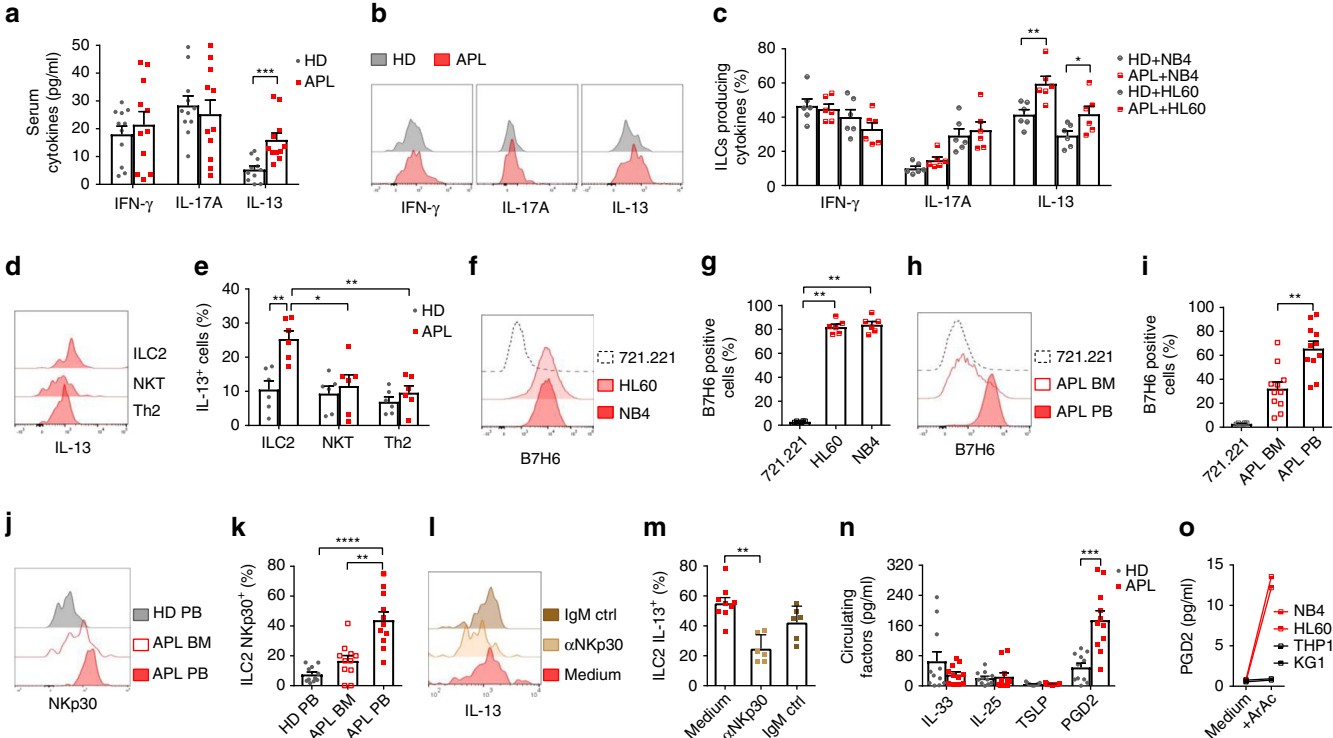

**Fig. 2** NKp30-B7H6 interaction and tumour-derived PGD2 favour IL-13 secretion. **a** Cytokine concentrations in serum samples of HD and APL patients (APL) ($n = 11$). **b** Representative examples of flow cytometry analysis of cytokine production by ILC in HD and APL patients (APL). **c** Frequencies of cytokine producing ILC in HD and APL patients (APL), upon co-culture with APL cell lines for 48 h (six independent experiments). **d** Representative example of flow cytometry analysis of IL-13-positive cells upon co-culture with autologous APL blasts. **e** Frequencies of IL-13+ ILC2, NKT and Th2 cells ($n = 6$). **f** Expression of B7H6 and **g** frequency of B7H6-positive cells in APL cell lines (NB4, HL60) and a control cell line (721.221, lymhphoblastoid cell line) (six independent experiments). **h** Representative example of flow cytometry analysis of the expression of B7H6 and **i** frequency of B7H6-positive cells in APL blasts in bone marrow (APL BM), in APL blasts in peripheral blood (APL PB), ($n = 11$) or in a control cell line (721.221, $n = 6$). **j** Representative example of flow cytometry analysis of the expression of NKp30 on ILC2 in peripheral blood of healthy donors (HD PB), in APL BM or in APL PB. **k** Relative frequencies of NKp30 expressing ILC2 in HD PB, APL BM or APL PB ($n = 11$). **l** Representative example of flow cytometry analysis of IL-13 produced by ILC2 co-cultured with the APL cell line NB4 in the presence of an anti-NKp30 blocking antibody (aNKp30), Ig control or medium. **m** Frequency of ILC2 IL-13+ in medium ($n = 9$), in the presence of an aNKp30 or an Ig control (IgM Ctrl) ($n = 6$, three independent experiments). **n** Quantification of IL-33, IL-25, TSLP and PGD2 in the sera of HD and APL patients (APL) ($n = 11$). **o** Quantification of PGD2 in supernatants of leukaemic AML (KG1, THP1) and APL cell lines (NB4, HL60), in the absence or presence of arachidonic acid (ArAc) (1 experiment). Error bars are s.e.m. Statistical analysis was performed using Mann-Whitney test (**a–c**, **n**) and Kruskal-Wallis test (**d–i**, **m**) and ANOVA test (**k**)

Since ILC2s can be activated by alarmins and PGD2, we quantified these mediators in the sera of APL patients at diagnosis. We found no detectable difference in IL-33, IL-25 or TSLP concentrations, but PGD2 concentrations were significantly elevated in APL patients compared to healthy donors (Fig. 2n). To test whether leukaemic cells may be the source of PGD2 in APL, we cultured the cell lines NB4 and HL60, and the AML cell lines KG1 and THP1 in presence of arachidonic acid. Only NB4 and HL60 cell lines produced PGD2 (Fig. 2o). Next, we asked whether PGD2 could further enhance IL-13 production by ILC2 in the presence of APL cell lines. We found that although PGD2 alone can induce IL-13 production in ILC2s, in agreement with prior reports[9], addition of PGD2 further enhances expression of IL-13 by ILC2s in combination with an APL cell line (Supplementary Fig. 2c).

To further verify the consistency of our observations, we retrieved mRNA expression data from The Cancer Genome Atlas (TCGA) for treatment-naive AML patients, including APL. TCGA data confirmed the specific upregulation of PGD synthase (but not PGE and PGI2), CRTH2 as well as B7H6 (but no other B7H molecules, except B7H3) in APL as compared to the other AML subtypes (Supplementary Fig. 2d).

Together, these findings suggest that circulating ILC2s in APL patients are primed to secrete IL-13 in response to malignant cells.

**ILC2-derived IL-13 promotes functional M-MDSC.** To understand the circuitries established by ILC2s through the secretion of IL-13, we screened different cell populations from APL patients and healthy donors for the expression of IL-13Rα1. Strikingly, CD14+ cells from both APL patients and healthy donors expressed high levels of this receptor (Fig. 3a, b). Since IL-13 is an inducer and activator of M-MDSCs, at least in mice[35, 54], and since IL-13 was upregulated in patients, we asked whether the CD14+ compartment of APL patients was enriched in M-MDSCs. As assessed by the expression of surface markers (Fig. 3c) and mRNA levels of arginase-1 and iNOS (Fig. 3d, e), CD14+ cells of APL patients were strongly enriched in M-MDSCs, and were present at increased frequencies compared to healthy donors (Fig. 3f). To further substantiate that ILC2s have direct effects on M-MDSC induction via IL-13, ILC2s were purified, expanded (Supplementary Fig. 3) and then stimulated with PGD2 and the NB4 APL cell line to secrete IL-13. Cell-free supernatants of these co-cultures were transferred onto purified CD14+ cells from

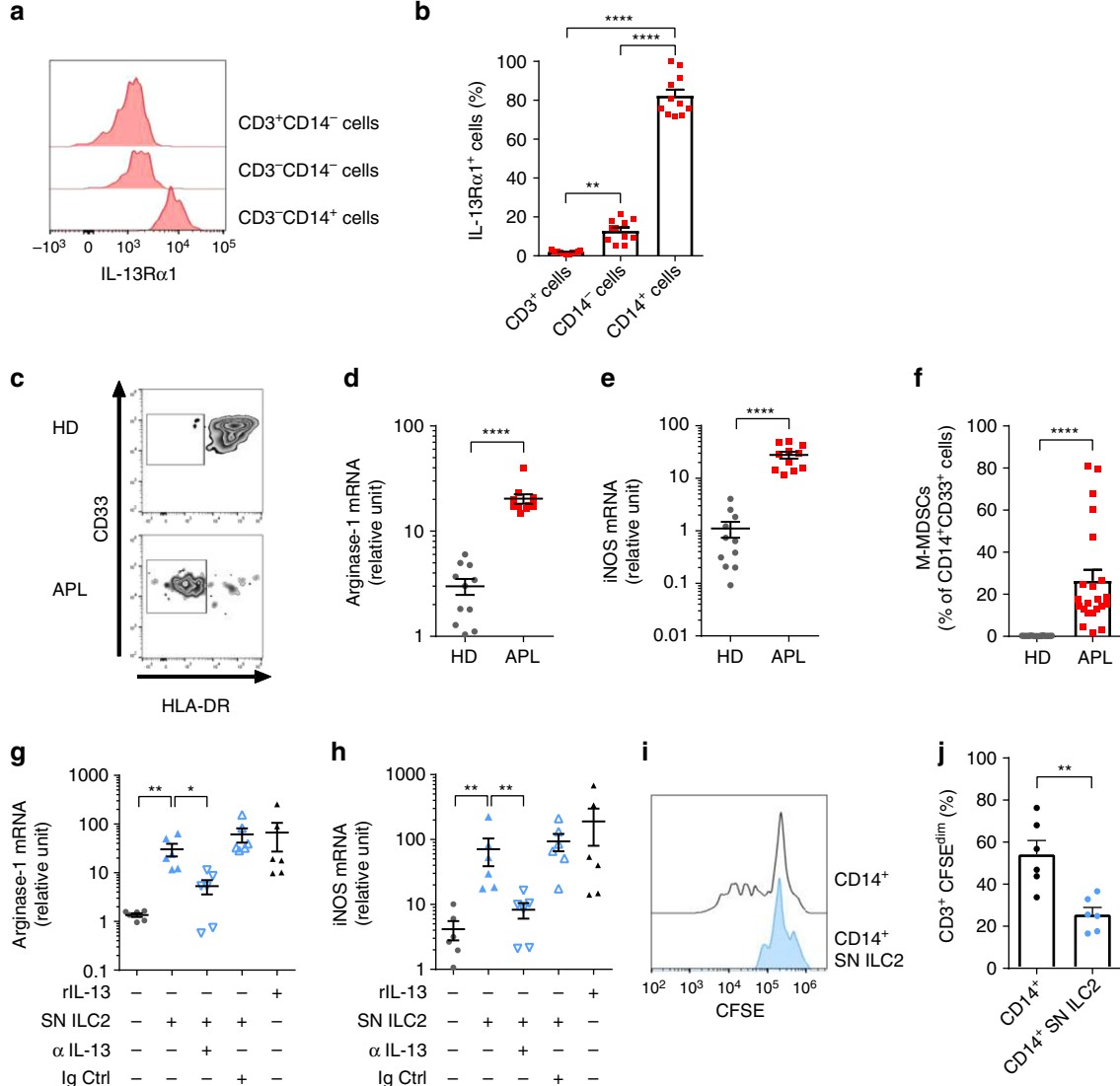

**Fig. 3** ILC2-derived IL-13 expands and activates suppressive M-MDSCs in APL. **a** Representative example of flow cytometry analysis of cell surface expression of the IL-13Rα1 on CD3[+], CD14[−], CD14[+] cells in peripheral blood of an APL patient. **b** Frequencies of IL-13Rα1 expressing cells in the different compartments (CD3[+], CD14[−], CD14 + $n$ = 11). **c** Representative example of flow cytometry phenotypic analysis of CD14[+] cells in healthy donors (HD) and APL patients (APL). Evaluated are the expression of HLA-DR and CD33, to define phenotypic M-MDSC. **d** Expression of arginase-1 and **e** iNOS in purified CD14[+] cells from HD and APL, as assessed by qPCR ($n$ = 11). **f** Frequency of M-MDSCs in peripheral blood of HD and APL ($n$ = 22). **g**, **h** Expression of arginase-1 (**g**) and iNOS (**h**) as assessed by qPCR, in purified CD14[+] cells cultured in medium supplemented or not with recombinant IL-13 (rIL-13), or cultured in supernatants derived from ILC2 cell lines, activated by PDG2 and the NB4 APL cell line (SN ILC2). A blocking anti-IL-13 antibody (aIL-13) or an Ig control (Ig Ctrl) were added where indicated ($n$ = 6, three independent experiments). **i** Representative example of CFSE-dilutions of CD3[+] T cells co-cultured with CD14[+] monocytes, or with in vitro-induced M-MDSCs (CD14[+]SN ILC2). **j** Frequencies of proliferating CD3[+] T cells upon co-culture with CD14[+] cells or with in vitro induced M-MDSCs (CD14[+]SN ILC2) ($n$ = 6, three independent experiments). *Error bars* are s.e.m. Statistical analysis was performed using ANOVA test (**b**), $t$ test (**d**–**f**), Kruskal-Wallis test (**g**, **h**) and Mann-Whitney test (**j**)

healthy donors. As shown in Fig. 3g, h, CD14[+] cells cultured with ILC2-derived supernatants upregulated arginase-1 and iNOS, which were abrogated by an anti-IL-13 blocking antibody. Importantly, ILC2 supernatant-induced M-MDSCs could attenuate autologous T-cell proliferation induced by anti-CD3/CD28 activation, indicating a bona fide suppressor cell functional phenotype (Fig. 3i, j).

These findings suggest that ILC2 drive the expansion of functional M-MDSCs in APL patients via IL-13.

**ATRA treatment reverses the increase of ILC2-MDSC in APL.** Collectively, the above findings suggest a scenario in which APL

blasts trigger the secretion of IL-13 by ILC2s through the combined engagement of CRTH2 by PGD2 and of NKp30 by B7H6, leading to the expansion and activation of M-MDSCs (Fig. 4a). To test this hypothesis, we compared each one of the components of this immunosuppressive axis in APL patients at diagnosis and in complete remission, upon disappearance of the APL blasts following ATRA treatment. In primary APL patients' samples collected after ATRA treatment, we found normalized levels of circulating PGD2, normalized presence of NKp30[+] ILC2s and of total ILC2s, and normalized levels of circulating IL-13 and M-MDSCs (Fig. 4b–f). In agreement with our in vitro data showing attenuated T-cell proliferation in the presence of M-MDSCs (Fig. 3i, j), CD8[+] T cells from treatment-naive APL

patients had lower Ki-67 expression and produced less granzyme B than T cells from patients in remission after ATRA therapy (Supplementary Fig. 4a, b). These findings suggest that complete remission was also accompanied by a recovery of T-cell functions.

To further explore the dynamics of the immunosuppressive axis that we have identified in patients, we used a transplantable mouse model of APL known to accurately recapitulate the human disease, including the sensitivity to ATRA[26, 27]. In this model, the human t(15;17) translocation that characterizes APL and gives

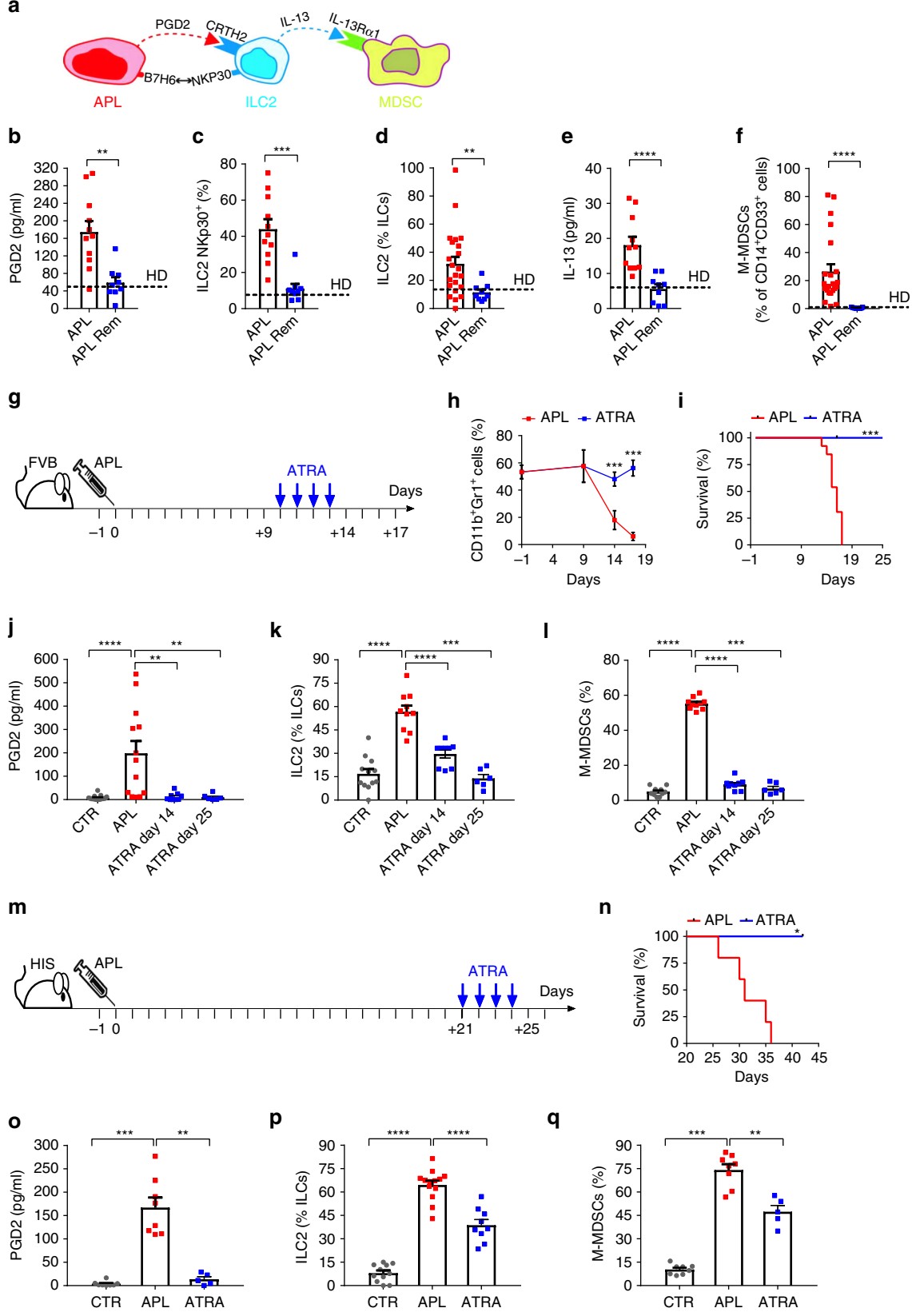

rise to the PML-RARA fusion protein is mimicked by the expression of the *PML-RARA* transgene in myeloid cells (referred to as PML-RARA APL mice; Fig. 4g–i). Importantly, in PML-RARA mice, the establishment of the disease resulted in increased levels of PGD2, ILC2s and bona fide M-MDSCs (Fig. 4j–l). Furthermore, achievement of remission following ATRA treatment was accompanied with normalized levels of PGD2, ILC2s and M-MDSCs as well as with a recovery of activated CD8+ T cells (Fig. 4j–l and Supplementary Fig. 4c–e). To determine if T cells played a major role in controlling APL engraftment, we moved to a less aggressive APL mouse model (i.e., APL B6) and we depleted CD4+ and CD8+ T cells. T-depleted APL B6 mice showed a shorter survival and an enhancement in ILC2s and M-MDSCs (Supplementary Fig. 4f–h).

Moreover, to be closer to the human immune system, we developed a mouse model in which immunodeficient NSG mice were reconstituted with a human immune system (HIS) and then injected with a human APL cell line (referred to as HIS APL mice), to test whether human APL cells had a direct effect on ILC2 expansion in vivo. APL establishment and clearance were followed by measuring luciferase intensity (Supplementary Fig. 4i, j). We observed an enhancement in PGD2 concentration and a major accumulation of ILC2s and M-MDSCs upon leukaemia engraftment that were significantly reverted by ATRA therapy (Fig. 4m–q). Importantly, ILC2 and Th2 were able to produce IL-13 upon in vitro stimulation, but only IL-13 production by ILC2 was inhibited by ATRA therapy in vivo (Supplementary Fig. 4k), while NKT cells were barely detectable. By monitoring cultured ILC2s exposed to ATRA, as well as in vivo ILCs in control HIS mice upon ATRA treatment we excluded a direct effect of ATRA on ILC2 (Supplementary Fig. 4l, m), although it was previously reported by others that ATRA might have a direct effect on ILC2s, at least in the gut[28].

These findings strongly support the sequence of events initiated by APL cells in vivo leading to expansion of highly suppressive M-MDSCs. Thus, this is a novel, ILC2-mediated, tumour immunosuppressive axis operating both in APL patients and in APL mouse models.

**PGD2/NKp30/ILC2/IL-13 blockade reverts immunosuppression**. To test our hypothesis that ILC2 play a central role in mediating increased M-MDSC levels upon APL progression, we used the previously described iCOS-T mouse model[29], to temporarily deplete ILC2s, while maintaining all the T-cell subsets, by the administration of diphtheria toxin (DTx, Fig. 5a). Also in this model, APL establishment resulted in ILC2 increase that was inhibited in the DTx-treated mice (Fig. 5b). Importantly, ILC2-depleted APL mice showed decreased M-MDSC levels, confirming a direct link between ILC2 and M-MDSC expansion (Fig. 5c).

According to our results, the binding of PGD2 to CRTH2 and B7H6 to NKp30 on ILC2 are the initial triggers of the ILC2-IL-13-M-MDSC immunosuppressive axis in APL. Therefore, to verify our hypothesis and identify druggable therapeutic targets in a suitable model, we aimed at blocking the prostaglandin metabolism in vivo. To do so, PML-RARA APL mice were treated once a day with indomethacin, a broad inhibitor of cyclooxygenase (COX) enzymes. As expected, indomethacin treatment resulted in low levels of PGD2. However, this treatment had no effect on the increase of ILC2s, most probably because by blocking COX, indomethacin not only inhibits PGD2, but also PGI2 synthesis, that was recently shown to downregulate the frequency and the function of ILC2s[12]. As a consequence of sustained ILC2 levels, M-MDSCs in indomethacin-treated mice remained as high as in untreated animals (Supplementary Fig. 5a–d). Therefore, we specifically blocked the PGD2 pathway by treating PML-RARA APL mice twice a day with TM30089, a specific antagonist of the PGD2 receptor. Following this treatment, PML-RARA APL mice showed low levels of PGD2 and a decrease in ILC2s and M-MDSCs (Supplementary Fig. 5e–h), demonstrating that the ILC2-M-MDSC immunosuppressive axis is partly driven by high PGD2 concentrations acting upon the CRTH2 receptor on ILC2s.

Since NKp30 does not exist in mice, we then aimed at blocking both the upstream and the downstream signals (i.e., PGD2 and IL-13) in PML-RARA APL mice. This combined treatment resulted in a decrease in PGD2 concentration, ILC2s, M-MDSCs and in a partial delay in mortality (Fig. 5d–h). Therefore, we hypothesised that the second signal coming by the APL cells (i.e., the binding of B7H6 to NKp30) was important in establishing/sustaining the immunosuppressive axis. Thus, the only setting where to test this hypothesis is offered by the APL HIS mice (in which NKp30 is expressed). In this model, we interfered with all different signals of the axis by using the PGD2 inhibitor, a human NKp30 blocking antibody, and an anti-human IL-13 neutralizing antibody (referred to as COMBO) (Fig. 5i). Following COMBO treatment, APL HIS mice showed reduced APL cell engraftment as well as a reduction in PGD2, ILC2s and M-MDSCs. This phenotype was accompanied with a significantly prolonged survival (Fig. 5j–o).

Taken together, our in vivo observations show that disruption of the novel tumour immunosuppressive axis by specifically blocking PGD2, IL-13 and NKp30 at least partially restores immunity resulting in increased survival in humanized leukaemic mice.

**ILC2s and M-MDSCs are enriched in prostate cancer**. To test whether this new immunosuppressive axis was established in other human tumours, we screened human cell lines derived from different solid tumours for their ability to secrete PGD2. Among

**Fig. 4** ATRA reverses the PGD2-ILC2-IL-13-M-MDSC immunosuppressive axis. **a** Schematic representation of the hypothetical immunosuppressive chain established in APL patients at diagnosis. **b** Comparison of serum concentrations of PGD2 (APL = 11; Rem = 9), **c** frequencies of ILC2 NKp30+ (APL = 11, APL Rem = 9), **d** frequencies of total ILC2 (APL = 22; APL Rem n = 9), **e** serum concentrations of IL-13 (APL n = 11; APL Rem n = 9) and **f** frequencies of M-MDSC (APL n = 22; APL Rem = 9) in peripheral blood of APL patients at diagnosis (APL) or in remission after ATRA treatment (APL Rem). *Dashed line*s represent mean values of all the parameters in HD. **g** Schematic representation of APL establishment and ATRA treatment schedule in FVB/NJ mice upon injection of APL splenocytes (three independent experiments), as assessed by quantification of the normal composition of the bone marrow (Gr1+CD11b+ cells) **h**, as previously described[27]. **i** Survival curves of untreated (APL, n = 13) and ATRA-treated (ATRA) APL mice (n = 6). **j** Quantification of PGD2 concentrations (CTR n = 15, APL n = 13, ATRA day 14 n = 9, ATRA day 25 n = 6), **k** ILC2 frequencies and **l** M-MDSCs frequencies in control FVB/NJ mice (CTR), APL mice before (APL) and after ATRA treatment (ATRA day14; ATRA day25) (CTR n = 12, APL n = 10, ATRA day 14 n = 9, ATRA day 25 n = 6). **m** Schematic representation of APL establishment and ATRA treatment schedule in HIS mice upon injection of an APL cell line (two independent experiments). **n** Survival curves of untreated (APL, n = 5) and ATRA-treated (ATRA, n = 3) APL HIS mice. **o** Quantification of PGD2 concentrations (CTR n = 8, APL n = 8, ATRA n = 5), **p** ILC2 frequencies (CTR n = 11, APL n = 13, ATRA n = 9) and **q** M-MDSCs frequencies (CTR n = 8, APL n = 8, ATRA n = 5) in APL HIS mice before (APL) and after ATRA treatment at day 25 (ATRA) (up to two independent experiments). *Error bars* are s.e.m. Statistical analysis was performed using Mann-Whitney test (**b–f**), Kruskal-Wallis test (**h**, **j–l**, **o–q**) and Log-rank test (Mantel-Cox) (**i**, **n**)

several cell lines, including colon, liver and kidney cancer cell lines, the established prostate cancer cell lines DU145 and PC3 produced the highest levels of PGD2 (Fig. 6a). Next, we assessed if these cell lines also expressed the NKp30 ligand B7H6. Similarly to APL blasts, prostate cancer cell lines expressed high levels of B7H6 (Fig. 6b). Then, we evaluated peripheral blood samples of patients suffering from prostate cancer for ILC subset distribution

and M-MDSC frequencies. Importantly, we found enrichment in ILC2s and NKp30[+] ILC2s in patients as compared to healthy donors (Fig. 6c, d) as well as increased frequencies of M-MDSCs, as shown in Fig. 6e. In periphery, no correlation was found between ILC2 and M-MDSC. However, ILC2 enrichment was stage dependent (Fig. 6f) and was not apparent in samples from patients with benign prostate hypertrophy.

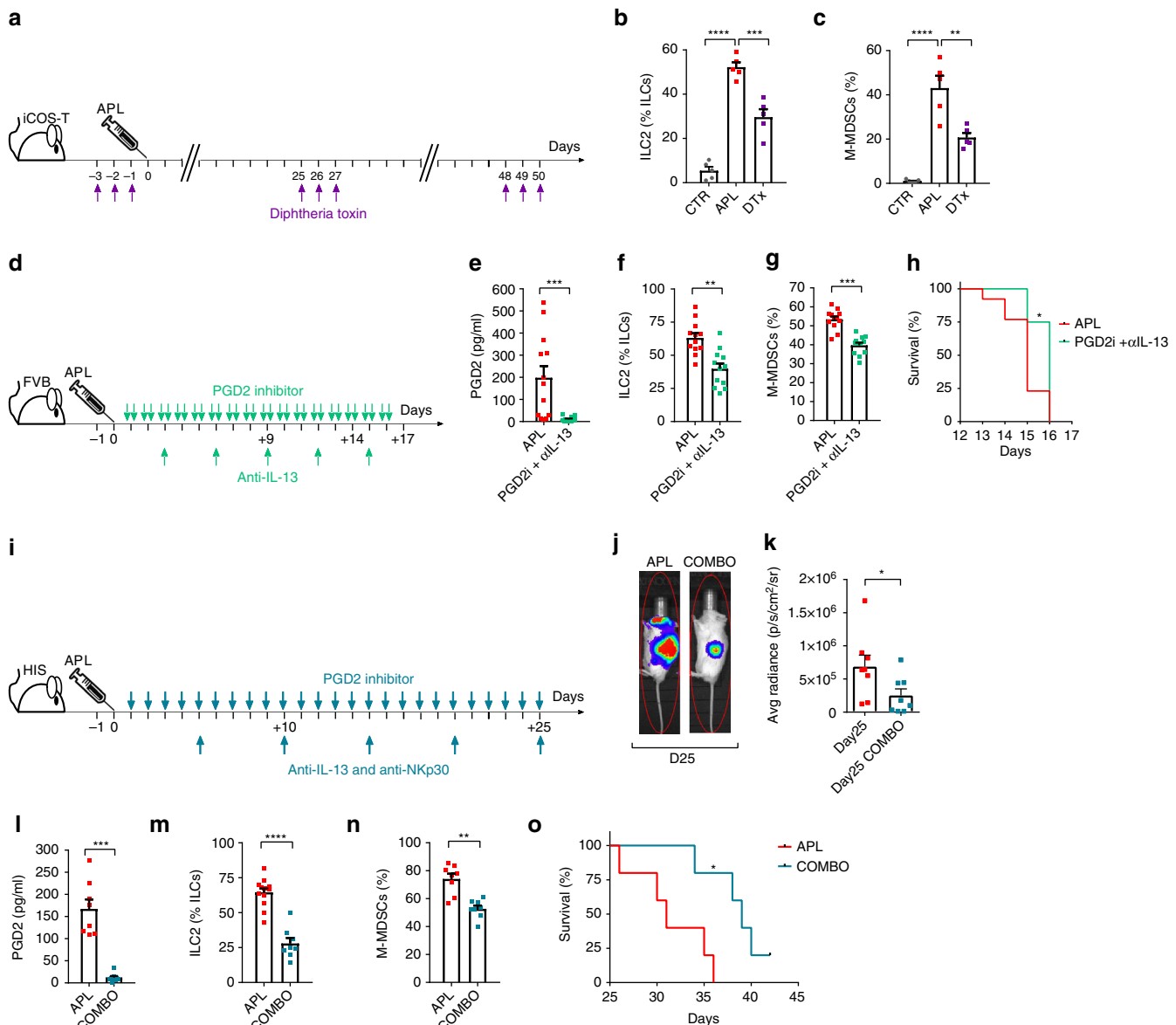

**Fig. 5** Therapeutic interference of the PGD2-ILC2-IL-13-M-MDSC immunosuppressive axis. **a** Schematic representation of DTx treatment schedule in iCOS-T mice upon injection of APL splenocytes (one independent experiment). **b** Quantification of ILC2 and **c** M-MDSCs frequencies in control mice (CTRL), APL mice (APL) and iCOS-T APL mice (DTx) ($n = 5$). **d** Schematic representation of the schedule of PGD2 inhibitor (TM30089) and anti-IL-13 neutralizing antibody treatment in FVB/NJ mice injected with APL blasts (two independent experiments). **e** Comparison of serum concentrations of PGD2 (APL $n = 13$, PGD2i + αIL-13 $n = 12$), **f** ILC2 frequencies (APL $n = 12$, PGD2i + αIL-13 $n = 12$), **g** M-MDSC frequencies (APL $n = 12$, PGD2i + aIL-13 $n = 12$). **h** Survival curves of untreated (APL) and PGD2i + αIL-13-treated FVB/NJ mice injected with APL blasts (APL $n = 12$, PGD2i + αIL-13 $n = 12$). **i** Schematic representation of the schedule of PGD2 inhibitor (TM30089), anti-NKp30 blocking antibody and anti-IL-13 neutralizing antibody treatment in HIS mice injected with APL blasts (one experiment). **j** Representative examples and **k** cumulative data of luminescence measurement in untreated APL HIS mice (APL) and mice treated with PGD2 inhibitor (TM30089), anti-NKp30 blocking antibody and anti-IL-13 neutralizing antibody (COMBO) for quantification of leukaemic engraftment ($n = 8$). **l** Quantification of PGD2 concentrations ($n = 8$), **m** ILC2 frequencies (APL $n = 13$, COMBO $n = 8$) and **n** M-MDSCs frequencies in untreated APL HIS mice (APL) and mice treated with PGD2 inhibitor (TM30089), anti-NKp30 blocking antibody and anti-IL-13 neutralizing antibody (COMBO) ($n = 8$). **o** Survival curves of untreated APL HIS mice and APL HIS mice treated with PGD2 inhibitor (TM30089), anti-NKp30 blocking antibody and anti-IL-13 neutralizing antibody (COMBO; $n = 5$). *Error bar*s are s.e.m. Statistical analysis was performed using Kruskal-Wallis test (**b**, **c**), Mann-Whitney test (**e-g**, **k**, **l-n**) and Log-rank test (Mantel-Cox) (**h**, **o**)

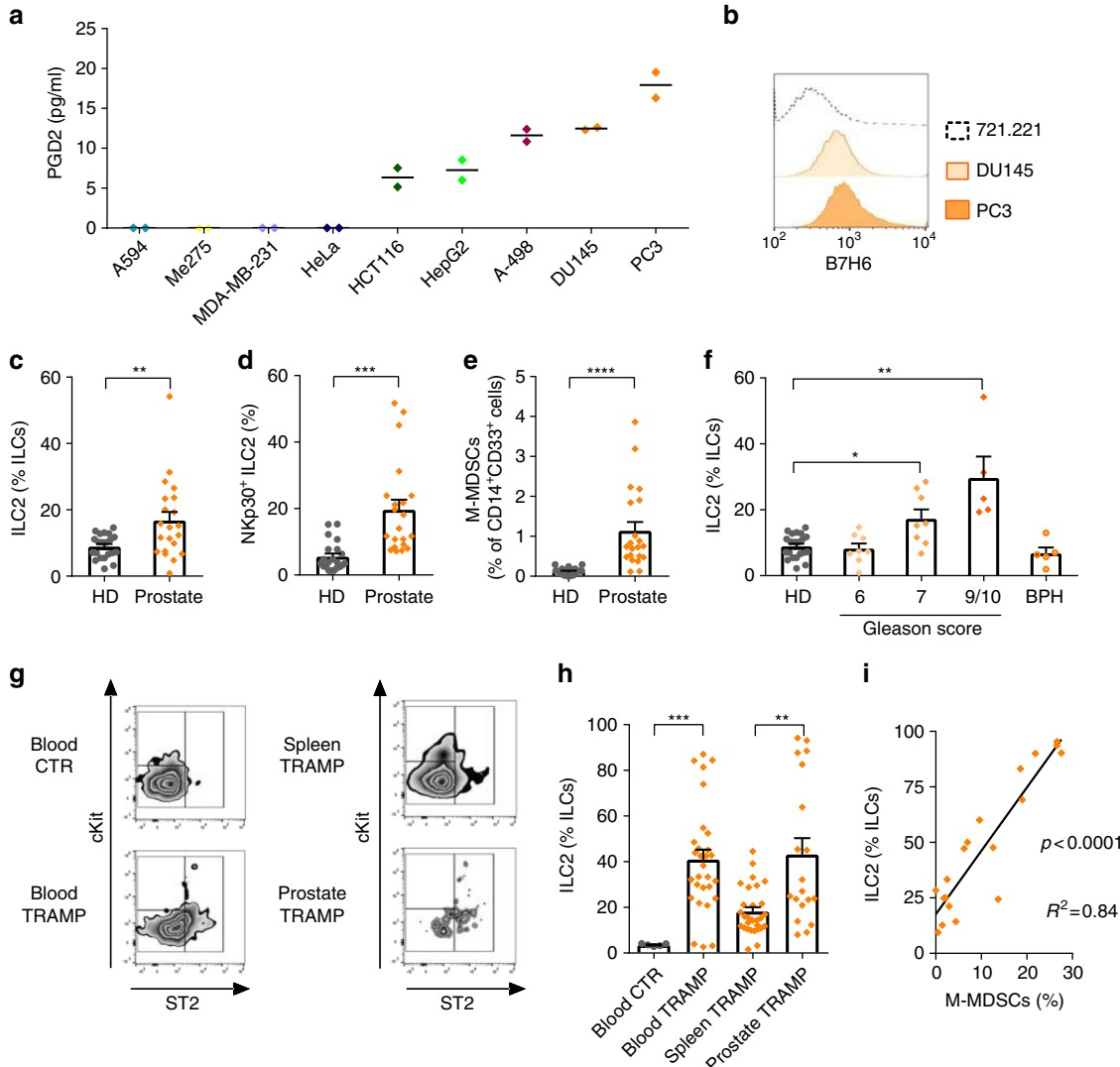

**Fig. 6** ILC2 are significantly increased in prostate cancer. **a** Quantification of PGD2 in supernatants of different human cancer cell lines ($n = 2$).
**b** Representative example of flow cytometry analysis of expression of B7H6 in prostate cancer cell lines DU145 and PC3, or in a control cell line (721.221).
**c** Relative frequencies of the ILC2 subset among total ILCs in healthy donors (*HD*) and prostate cancer patients (Prostate) ($n = 21$). **d** Relative frequencies of NKp30 expressing ILC2 in peripheral blood of HD and Prostate ($n = 21$). **e** Frequency of M-MDSC in peripheral blood of HD and Prostate ($n = 21$).
**f** Relative frequencies of the ILC2 subset among total ILCs in healthy donors (HD, $n = 21$) and prostate cancer patients at stage (Gleason score) 6 ($n = 8$),
7 ($n = 8$), or 9/10 ($n = 5$) or in patients with benign prostate hypertrophy (*BPH*, $n = 5$). **g** Representative examples of flow cytometry analysis of innate
lymphoid cell subsets in blood of FVB female mice (blood CTR, $n = 5$) and in blood ($n = 29$), spleen ($n = 29$) and prostate ($n = 19$) of TRAMP mice.
**h** Relative frequencies of the ILC2 subset among total ILCs in blood of FVB female mice (blood CTR) and in blood, spleen and prostate of TRAMP mice.
**i** Correlation of ILC2 and M-MDSCs in prostate tumours of TRAMP mice ($n = 19$) (three independent experiments). *Error bars* are s.e.m. Statistical analysis
was performed using *t* test (**c**–**e**, **h**) and Kruskal-Wallis test (**f**)

Given the inaccessibility to human prostate tumour specimens, we used a mouse model in which mice uniformly and spontaneously develop autochthonous (orthotopic) prostate tumours following the onset of puberty (referred to as TRAMP mice)[30]. In agreement with our observations in patients, we found that ILC2s were increased in the blood of TRAMP mice as compared to the blood of female mice of the same strain (FVB). Notably, in paired spleen and prostate samples of TRAMP mice, ILC2s were highly enriched within the prostate tumours (Fig. 6g, h). Thus, we assessed the presence of M-MDSCs in the same tumours and observed that M-MDSCs frequencies positively correlated with tumour-infiltrating ILC2s (Fig. 6i). In addition, we quantified T, NKT, B cells, as well as eosinophils in the prostate, blood and spleen of tumour-free and tumour-bearing TRAMP mice (i.e., before and after 12 weeks of age) and we

found no difference in the blood and in the spleen. However, in the prostate, beside the enhancement in M-MDSCs already described above, we found a reduction of NKT cells (Supplementary Fig. 6a–c).

Overall, these data in prostate cancer, although only correlative, suggest that the expansion in M-MDSCs within the tumour might be driven by the increase in ILC2s responding to tumour-secreted PGD2.

## Discussion

In this study, by combining the analysis of primary APL patients' samples and in vivo APL models, we have uncovered a novel tumour immunosuppressive axis initiated by malignant cells. Our data are consistent with a sequence of events driven by

tumour-derived PGD2 associated with engagement of the NKp30-B7H6 pathway leading to significant ILC2 activation and expansion. In turn, increased IL-13 secretion by primed ILC2s results in a significant expansion of activated M-MDSCs, which may inhibit anti-tumour immune responses. We show that in vivo interference with PGD2, NKp30 and IL-13 functions in murine and humanized APL models is sufficient to significantly reverse immunosuppression and increase survival of humanized tumour-bearing mice. These data demonstrate unprecedented ILC2-mediated tumour immunosuppression in human malignancies.

Arachidonic acid catabolism by cyclooxygenase enzymes and specific synthases is dysregulated in a number of tumours, including pancreatic, lung, breast, colon, prostate and bladder cancer, due either to the overexpression of cyclooxygenases and prostaglandin synthase enzymes[23, 31–33], or by dysfunctions of the prostaglandin degrading enzymes, leading to an accumulation of intermediate metabolites[34]. Recently, some members of the PG family, namely PGI2, PGD2 and PGE2, have emerged as key regulators of ILC2 and ILC3 functions and systemic inflammation[9, 12, 35]. Yet, nothing is known on the role of PG on ILC functions in anti-tumour immunity. We document for the first time a correlation between tumour-derived PGD2 and ILC2 accumulation in cancer patients. The observation, that CD4[+] Th2 and NKT cells are unaltered in APL patients compared to controls, suggests that PGD2 signalling acts on ILC2s in a T-cell-independent manner. In that regard, the increased expression of NKp30 on patients' ILC2s and of B7H6 on malignant APL blasts might act as second indispensable signal for a specific ILC2 activation. NCR engagement was already shown to be crucial in the context of human lung cancer, where the expression of another NCR, NKp44, by ILC3 NCR[+] was shown to promote formation of tertiary lymphoid structures[36]. Here in contrast to the protective role of NKp44 ligation on ILC3, NKp30 triggering on APL patients' ILC2s favours the release of large quantities of IL-13 that has been recently shown, at least in mice, to promote M-MDSC activation and expansion[35, 54].

Elevated serum IL-13 concentrations correlating with increased M-MDSC levels have been reported in patients with different solid tumours[37]. Yet, the source of IL-13 was considered to be CD4[+] Th2 cells or malignant cells. This paradigm has recently been revisited in the context of type 2 immunity in infectious diseases, asthma and skin autoimmunity, where ILC2s appear to be the initial source of IL-13, acting as indispensable contributors for efficient pathogen clearance[29, 38] and pathology development[14], respectively. In the context of APL, we show that antigen-independent IL-13 secretion by ILC2s is pivotal for the functionality of M-MDSCs[39]. Moreover, in haematologic malignancies, IL-13 was reported to also exert pro-tumoural activity on mesenchymal stromal cells[40], an observation that warrants further verification in other types of tumours.

M-MDSCs may exert their immunosuppressive functions through different mechanisms. They were shown to promote induction of regulatory T cells (Treg)[41], however this component does not seem to be the predominant one in APL, since Treg levels in patients were normal (Supplementary Fig. 2a). Rather, we speculate that by iNOS and arginase-1 upregulation, M-MDSCs act on effector cells, by inhibiting T and NK cell functions, thus impairing efficient leukaemic cell clearance[42].

The unique diagnostic and therapeutic setting of APL allowed us to follow the newly identified immunosuppressive axis in patients from disease onset to complete remission. In agreement with our hypothesis, ATRA-induced differentiation and eventual death of malignant APL blasts resulted in a normalization of PGD2 levels, as well as a complete restoration of all downstream mediators of the pathway (i.e., ILC2s, NKp30, IL-13, M-MDSCs,

T-cell effector functions). In our APL mouse models, beside the use of ATRA, we tested a broad blocker of COX (indomethacin), as well as a specific PGD2 inhibitor (TM30089). Only the latter showed a partial resolution of the immunosuppressive axis, probably due to the fact that other PG metabolites exert opposite effects on specific ILC subsets. Indeed, it was recently reported that PGI2 inhibits ILC2s, while PGE2 directly acts on ILC3s by promoting their anti-inflammatory function[12, 35]. As further proof of the involvement of IL-13-producing NKp30[+]ILC2s in directly controlling activation and expansion of M-MDSCs in our models, we used temporally ILC2-depleted iCOS-T mice injected with leukaemic cells and showed that a loss in ILC2s results in a decrease of M-MDSCs. To be closer to a therapeutic setting in patients, where ILC2 depletion might not be a feasible option, we combined inhibition of PGD2 with NKp30 and IL-13 blocking antibodies. The treatment resulted in partial normalization of the axis and increased survival in both models used. Due to the lack of NKp30 expression in mice, it will be highly important to further test the clinical relevance of our observations in NSG and HIS mice transferred with human APL cell line expressing PML-RARA fusion protein or human primary APL blasts, to provide additional mechanistic clues. However, in contrast to other AML subtypes, engraftment of primary APL is documented to be poor[43]. Once established, these models would be valuable tools for future in vivo testing of treatment options of ILC2 modulators, for rapid translation to the clinics.

In that context, the importance of therapeutically correcting immunosuppressive barriers in cancer patients has been recently highlighted by the impressive clinical results of immune check-point blockade using monoclonal antibodies against CTLA-4 and PD1/PDL1[44]. Thus, the novel immunosuppressive axis that we characterized in APL might be targetable in patients with solid tumours of different histological types for which the benefit of current therapies is still unsatisfactory. Indeed, in human tumours with altered arachidonic acid catabolism or exaggerated alarmin production (e.g., IL-33[45]), the ILC2-IL-13-M-MDSC network might represent a novel, early acting innate pathway of immune regulation.

In conclusion, we show that ILC2s may be considered a novel modulator of tumour immunosuppressive networks that can connect signals derived from tumour cells with tolerogenic cells that facilitate tumour growth.

## Methods

**Human cell collection.** Venous blood was drawn from healthy donors at the local Blood Transfusion Center, Lausanne, Switzerland, under the approval of the Lausanne University Hospital's Institute Review Board. Peripheral blood and BM samples were obtained from patients with PML-RARα-positive acute promyelocytic leukaemia, at diagnosis ($n = 22$) or during remission after ATRA treatment ($n = 9$), from treatment-naive patients with prostate adenocarcinoma ($n = 21$) and benign prostate hypertrophy ($n = 5$) at clinical centres in Bologna, Milano, Rozzano, Lausanne, Bern, Bergamo and Padova. The study was approved by the Institutional Review Boards of the University Hospital of Bologna, the San Raffaele Hospital, the Humanitas Research Hospital, the Lausanne University Hospital, the Bern University Hospital, the Hospital of Bergamo and Padova (EC consents: EMATC-2013-01, EC 1720, EC 11-09-2006, APL01, 1237-25090 and 119/10). Written informed consent was obtained from all healthy subjects and patients, in accordance with the Declaration of Helsinki. Fresh anticoagulated blood diluted at 1:2 ratio in PBS was layered on lymphoprep (ratio diluted blood: lymphoprep 1.5:1). Mononuclear cells were isolated by density gradient centrifugation (1800 rpm, 20 min centrifugation without break, room temperature), washed and immediately cryopreserved in 50% RPMI, 40% FCS and 10% DMSO. Serum samples were also collected at the same sampling day after centrifugation of whole blood at 2000 rpm for 10 min, at room temperature, and immediately frozen. Of the 22 APL patients, for in vitro and ex vivo assays, we selected samples exclusively according to cell viability (more than 70% living cells) and counts (more than $1 \times 10^6$ living cells) upon thawing.

**Cell culture.** Human leukaemic lines (KG1, catalogue number (cn): CCL-246, ATTC; THP1, cn: TIB-202, ATCC, HL60, cn: CCL-240, ATCC; NB4, cn: ACC207,

DSMZ)[26], lung adenocarcinoma (A549, cn: CCL-185, ATCC), malignant mela-noma (Me275, derived in-house, Ludwig Institute for Cancer Research, Lausanne Branch, Lausanne, Switzerland), breast (MDA-MB-231, cn: CRM-HTB-26, ATCC), ovarian (HeLa, cn: CCL-2, ATCC), colon (HCT116, cn: CCL-247, ATCC), liver (HepG2, cn: HB-8065, ATCC), kidney (A-498, cn: HTB-44, ATCC) and prostate cancer (DU145, cn: HTB-81, ATCC and PC3, cn: CRL-1435, ATCC) cell lines were owned by the host laboratory and maintained in tissue culture flasks in RPMI supplemented with 10% FCS, amino acids and HEPES. All cell lines were periodically tested for mycoplasma contamination and confirmed negative by PCR with mycoplasma-specific primers (5′-ACTCCTACGGGAGGCAGCAGTA-3′ and 5′-TGCACCATCTGTCACTCTGTTAACCTC-3′). When indicated, tumour cell lines were cultured at $1 \times 10^6$ cells/ml for 4 h in the presence of 30 µM arachidonic acid (Sigma).

Human peripheral blood mononuclear cells were cultured in RPMI supplemented with 8% heat-inactivated, pooled human serum, in the presence of 20 U/ml rhIL2 (Proleukin, Roche). When indicated, PGD2 (Sigma), IL-33 (Adipogen), ATRA (Sigma), anti-human NKp30 or anti-human IL-13 blocking antibodies, or Ig controls were added. Human ILC2 cell lines were established from PBMCs by flow cytometry-based sorting of Lin⁻, CD45⁺, CD127⁺, CRTH2⁺ and CD161⁺ after depletion of CD3⁺ (Microbeads, cn: 130-050-101), CD19⁺ (Microbeads, cn: 130-050-301), CD14⁺ (Microbeads, cn: 130-050-201) cells by AutoMACS (Supplementary Fig. 3). ILC2 cell lines were expanded in Yssel's medium supplemented with 1% AB human serum, with irradiated (dose: 30 Gy, source: CS-137) allogenic PBMCs and 100 U/ml rhIL2 for 3 weeks.

For in vitro experiments using CD14⁺ cells, pure ILC2 cell lines were co-cultured with the NB4 APL cell line at a 1:1 ratio and 100 nM PGD2 for 48 h. Cell-free supernatants were added to magnetically purified CD14⁺ cells maintained in 10 ng/ml GM-CSF, in the presence or absence of 10 µg/ml anti-IL-13 neutralizing antibody, an Ig control or recombinant human IL-13 (BioLegend) for 6 days, as indicated. cDNA was extracted from CD14⁺ and processed as described below.

For in vitro proliferation assays, CD14⁺ cells (human), cultured in the presence or absence of ILC2-conditioned supernatants, or CD11b⁺Gr1^dim (mouse) cells were co-incubated at 1:1 ratio with magnetically enriched CD3⁺ T cells, previously labelled with CFSE. Proliferation of CD3⁺ T cells stimulated with plate bound anti-CD3 and soluble anti-CD28 antibodies (1 µg/ml each) was assessed after 4 days by measurement of CFSE dilution by flow cytometry.

**APL model in FVB/NJ background.** Murine cells carrying the *PML/RARα* translocation (PML-RARA APL cells) were transplanted as reported[26, 42, 46]. Briefly, $15 \times 10^6$ live cells from spleens of animals with leukaemia were intrave-nously (i.v.) transferred in 200 µl PBS into previously irradiated (dose: 3,5 Gy, source: CS-137) FVB/NJ mice that are bred in-house and maintained into a barrier animal facility. For secondary transplantation, $1.5 \times 10^6$ live cells from primary recipients' spleens were i.v. transferred into non-irradiated syngeneic mice. In each experiment PML-RARA APL cells from a single primary recipient were used to generate all secondary recipients and secondary recipients were used for all treatments. ATRA (Sigma) (65 mg/kg body weight, i.p.) was administered from day 9 to 12. TM30089 (Cayman Chemicals) (5 mg/kg body weight) was administered orally, twice a day. Anti-IL-13 antibody (150 µg/mouse/dose) was administrated every 5 days intraperitoneally (i.p.). Indomethacin (Sigma) (2 mg/kg body weight) was orally administered once a day. Female mice between 7 and 15 weeks old kept in a barrier animal facility were randomly assigned to specific treatment groups, and the investigator was blinded to the group allocation. Animals were monitored three times per week during the experiment and clinical signs of disease were assessed by the presence of blast cells by May-Grünwald-Giemsa staining in blood smears. At the end of the experiment, mice were killed by $CO_2$ inhalation. This study was approved by the local Veterinary Authority (VD Service de la con-summation et des affaires Vétérinaires) under the licence VD1850 and performed in accordance with Swiss ethical guidelines.

**APL model in C57BL/6 background.** iCOS-T mice from 6 to 12 weeks old were kindly provided by Dr Andrew McKenzie from MRC-LMB in Cambridge (UK) and WT C57BL/6JOLaHsd matching-aged female mice were purchased from Envigo (stock number: 5704F). In order to deplete ILC2, iCOS-T mice were injected i.p. with 1 µg of diphtheria toxin (Sigma) for three consecutive days at days (i) −3, −2, −1, (ii) 25, 26, 27 and (iii) 48, 49, 50.

In a second cohort, WT mice were depleted from CD4⁺ and CD8⁺ T cells. Briefly, at day −2, mice were injected with 400 µg/mouse of anti-CD4 (GK1.5) and 200 µg/mouse anti-CD8 (53-6.7). Then anti-CD4 (200 µg/mouse/dose) and anti-CD8 (100 µg/mouse/dose) were injected every 5 days until the end of the experiment. At day 0, $1 \times 10^6$ APL cells were transferred i.v. into the tail vein; bone morrow was monitored at days −1, 29, 49 and at the end of the experiment. This experiment followed the local Veterinary Authority (VD Service de la consummation et des affaires Vétérinaires) under the licence VD1850 and Swiss ethical guidelines.

**TRAMP mouse model.** Blood, prostate and spleen were collected from 12- to 18-week-old male TRAMP-FVB/NJ mice[30] or control female FVB/NJ mice (blood

only). TRAMP-FVB/NJ mice are bread in-house and kept in in our barrier animal facility. For immune cell isolation, prostate tissue was cut into small pieces in dissociation medium containing 100 U/ml collagenase and 100 µl/ml DNAse. Samples were then incubated under agitation at 37 °C during 1 h 15 min. After filtration through a 0.70 µm nylon filter, samples were centrifuged and washed with PBS prior to staining. Animals were monitored three times per week during the experiment for clinical signs of disease starting at week 14. At the end of the experiment, mice were killed by $CO_2$ inhalation. This study was approved by the local Veterinary Authority (VD Service de la consummation et des affaires Vétérinaires) under the licence VD1850 and performed in accordance with Swiss ethical guidelines.

**Humanized mouse model.** Humanized mice (HIS mice) were generated in our laboratory. Briefly, immune-deficient NSG (NOD.Cg-Prkdc_scid Il2rg_tm1Wjl/SzJ) mice were injected intrahepatically with 100,000 human CD34⁺ hematopoietic stem cells (HSC) and analysed for human immune cell reconstitution after 12 weeks using multi-colour antibody panels established for immune-phenotyping of human whole blood samples as described[47], with some minor modifications to include ILC reconstitution (Lin⁻CD127⁺ cells). Once reconstituted, 12–14 weeks post trans-plantation, HIS mice, either males or females, were i.v. injected with enhanced luciferase[48] transfected HL60 APL cells previously tested for mycoplasma con-tamination ($2.5 \times 10^6$ live cells/mouse). ATRA (Sigma) (65 mg/kg body weight, i.p.) was administered daily from day 21 to 24. Hundred and fifty µg/mouse/dose of anti-NKp30 and anti-IL-13 were injected every 5 days; to monitor leukaemia progression and treatment effectiveness, mice were injected with luciferin (3 mg/mouse) at days 7, 17, 25. Images were taken in the Xenogen Ivis 100 in vivo Imaging System 20 min after the injection (60 s, Bin4, F/Stop 1,2). Animals were randomly assigned to specific treatment groups, and the investigator was blinded to the group allocation. Human CD34⁺ cells were purchased from either Lonza (CH) or AllCells (USA). NSG mice were purchased from The Jackson Laboratory (stock number: 005557) and bred, treated and maintained under pathogen-free conditions in-house. Animal experimentation followed protocols approved by the local Veterinary Authority (VD Service de la consummation et des affaires Vétérinaires) under the licence VD2797 and performed in accordance with Swiss ethical guidelines.

**Antibodies for flow cytometry on human cells.** Human ILCs were identified as Lineage (FITC-conjugated anti-CD3 (cn: A07746, 1:50), anti-CD4 (cn: A07750, 2:50), anti-CD14 (cn: B36297, 1:100), anti-CD16 (cn: B49215, 1:100), anti-CD19 (cn: 07768, 1:100) (Beckman Coulter), anti-CD8 (cn: MCA1226F, 1:50), anti-CD15 (cn: MCA2458F, 2:50) (AbD Serotech), anti-CD20 (cn: 302304, 1:100), anti-CD33 (cn: 303304, 1:50), anti-CD34 (cn: 343604, 1:100), anti-CD203c (cn: 324614, 1:50) and anti-FcεRI (cn: 334608, 2:50) (Biolegend)) negative CD127⁺ (PerCP-Cy5.5-(cn: 3151322, 1:100) or Brillant Violet 421-conjugated (cn: 351310, 1:100) anti-CD127 (BioLegend) lymphocytes. ILC subsets were identified using PE CF594- (cn: 563501, 1:100) or Brillant Violet 421-conjugated (cn: 562992, 2:50) anti-CRTH2 (BD Pharmingen); PerCP-Cy5.5- (cn: 331920, 1:100) or PE-Cy7-conjugated (cn: 562101, 1:100) anti-NKp46 (BioLegend, BD Pharmingen); PE- (cn: 313204, 1:100) or APC-conjugated (cn: 550412, 1:50) anti-cKit (BioLegend, BD Pharmingen); APC-eFluor780-conjugated anti-CD56 (cn: 47-0567-42, 2:50 eBioscience). Dead cells were always excluded using a ViViD LIVE/DEAD fixable dead cell stain kit (cn: L34957, LifeTechnologies) (Supplementary Fig. 1a). Where indicated, addi-tional markers were evaluated using a purified anti-B7H6 antibody (cn: ab121794, 1:100, Abcam) and an AlexaFluor647-conjugated donkey anti-rabbit secondary antibody (cn: A31573, 1:4000, Life Technologies), PE-conjugated anti-NKp30 (cn: 12-3379-42, 1:100, eBioscience), AlexaFluor647-conjugated anti-Ki-67 (cn: 350510, 1:100, BioLegend), FITC-conjugated anti-Bcl-2 (cn: 340575, 2:50, BD Pharmingen), FITC-conjugated anti-Granzyme B (cn: 515403, 2:50, BioLegend) antibodies, PE-conjugated anti-6B11 (cn: 552825, 4:50, BD Pharmingen), PE-Cy7-conjugated CD1a (cn: 25-0019, 1:50, eBioscience), APC-Cy7-conjugated anti-CD5 (cn: 364009, 1:100, BioLegend), PE-conjugated anti-human TCRαβ (cn: A39499, 2:50, Beckman Coulter) and PE-conjugated anti-TCRγδ (cn: 331209, 2:50, BioLegend).

Human M-MDSCs were identified using Pacific Blue-conjugated anti-CD14 (cn: 301816, 2:50, BioLegend), PerCP-Cy5.5-conjugated anti-CD15 (cn: 323020, 1:50, BioLegend), APCeFuor780-conjugated anti-CD11b (cn: 47-0118, 1:50, eBioscience), APC-conjugated anti-CD33 (cn: 303408, 1:50, BioLegend), PE-Cy7-conjugated anti-HLA-DR (cn: 307616, 2:50, BioLegend), PE-conjugated anti-IL-13Rα1 (cn: FAB1462P, 1:50, R&D Systems) and Lineage markers (PE-AF610-conjugated anti-CD3 (cn: MHCD0322 1:100), PE-AF610-conjugated anti-CD19 (cn: MHCD1922, 1:50) and PE-TexasRed-conjugated anti-CD56 (cn: MHCD5617, 1:100, all from Invitrogen).

For cytokine production, MNCs were stimulated ex vivo for 3 h with PMA/ionomycin or were co-cultured with APL cell lines overnight at a 1:1 ratio, in the presence or absence of PGD2, anti-NKp30 antibody (F525) or an IgM control. BrefeldinA was added at 2.5 µg/ml after 1 h of co-culture. Intracellular staining was performed using PE-Cy7-conjugated IFN-γ (cn: 557844, 1:200), APC-conjugated anti-IL-13 (cn: 561162, 1:100) (both from BD Pharmingen) and AlexaFluor700-conjugated anti-IL-17A (cn: 512318, 1:100, BioLegend) after fixation and permeabilization with 0.1% saponin (Sigma). Samples were acquired on a Gallios

flow cytometer (Beckman Coulter) and data were analysed using FlowJo software (TreeStar).

**Antibodies for flow cytometry on mouse cells.** Murine ILCs were identified as Lineage (FITC-conjugated anti-CD3e (cn: 130-102-207, 1:200), anti-CD5 (cn: 130-106-202, 1:200), anti-CD11b (cn: 130-098-085, 1:200), anti-CD11c (cn: 130-102-466, 1:200), anti-CD19 (cn: 130-102-494, 1:200), anti-Ter119 (cn: 130-102-257, 1:200), anti-B220 (cn: 130-102-228, 1:200), anti-TCRγδ (cn: 130-109-796, 1:200), anti-DX5 (cn: 130-102-258, 1:200), anti-FcεRI (cn: 130-102-264, 1:50) (all from Miltenyi Biotec)) negative, CD127$^+$ (eF660 anti-CD127 (clone: A7R34, 1:100, eBioscience)), CD45$^+$ (PE-Cy7-conjugated anti-CD45 (clone: 30F11, 1:300, Bio-Legend)) lymphocytes. ILC subsets were identified using PE-conjugated anti-ST2 (cn: 145304, 1:100, Biolegend); Brilliant Violet 421-conjugated anti-NKp46 (cn: 130-102-185, 1:50, Miltenyi Biotec); PerCP-eFluor710-conjugated anti-cKit (clone: 2B8, 1:200, eBioscience).

Murine M-MDSCs were identified using PE-Cy7-conjugated anti-CD45 (clone: 30F11, 1:300, BioLegend), AlexaFluor700-conjugated anti-CD11b (clone: M1/70, 1:300, eBioscience) APC-eFluor780-conjugated anti-Gr1 (clone: RB6 AC5, 1:500, BD Pharmingen), APC-eFluor780-conjugated anti-Ly6C (cn: 47-5932-80, 1:600, eBioscience), PerCP-Cy5.5-conjugated anti-Ly6G (cn: 45-5931-80, 1:600, eBioscience) and PE-Cy7-conjugated anti-CD49d (clone 9F10, 1:200, BioLegend) antibodies, as previously described[49, 50].

Additional antibodies used were Pacific Blue-conjugated anti-CD44 (clone: IM781, 1:100), Alexa647-conjugated anti-CD62L (clone: Mel14, 1:1000), TexasRed-conjugated anti-B220 (cn:551489, 1:50, BD Pharmingen) and Brilliant Violet 421-conjugated anti-SiglecF (cn: 565934, 1:200, BD Pharmingen) antibodies.

**cDNA synthesis and qRT-PCR.** Dry pellets of CD14$^+$ cells from healthy donors or APL patients and in vitro-induced M-MDSC were resuspended in a lysis/cDNA mix solution containing 6.3 µl of "lysis buffer" (prepared as previously described by Gupta et al.[51]), 3 µl of "5× RT buffer"[51], 1.5 µl of 0.1 M DTT (AppliChem), 0.75 µl of 10 mM dNTPs (Invitrogen), 0.25 µl of 100 ng/µl oligo-(dT) (Metabion), 0.4 µl of MMLV reverse transcriptase (Invitrogen), 0.2 µl of RNAsin (Promega) and 2.6 µl RNAse free water. To allow reverse transcription into cDNA, samples were incubated at 37 °C for 60 min. The transcriptase was then inactivated at 90 °C for 3 min and samples were stored at −80 °C. Quantitative real-time PCR was performed using KAPA SYBR® FAST master mix with ROX (KAPA Biosystems), 200 nM of each primer and 1/50 of the reverse transcription reaction. Samples were run at least in duplicate in a ABI 7500 Fast Real-Time PCR thermocycler (Applied Biosystems) with the following parameters: 3 min at 95 °C for enzyme activation, 40 cycles at 95 °C for 5 s and 60 °C for 30 s, followed by amplicon melting analysis to evaluate the specificity of the reaction and identify the presence of primer-dimers. Primers sequences were as follows: for GAPDH, 5′-GGACCTGACCCTGCCGTC TAG-3′ (forward) and 5′-CCACCACCCTGTTGCTGTAG-3′ (reverse); for arginase-1: 5′-ATTCTTCCGTTCTTCTTGACTT-3′ (forward) and 5′-AGTGTGATG TGAAGGATTATG-3′ (reverse); for iNOS: 5′-ATGCTCAGCTCATCCGCTAT-3′ (forward) and 5′-CACAAGGTCAGGTGGGATTT-3′ (reverse); and were purchased from Microsynth (Switzerland). Results were normalized to GAPDH and expressed as $2^{-\Delta Ct}$.

**ELISA.** Human IL-13 ELISA (cn: 435207, Biolegend), mouse IL-13 ELISA (cn: 88-7137-22, eBioscience), human IL-33 (cn: 435907, Biolegend), human IL-25 ELISA (cn: KA2190, Abnova), human TSLP ELISA (cn: 434207, Biolegend), PGD2-MOX ELISA (cn: 512011, Cayman Chemicals) were performed according to the manufacturer's instructions. Human IL-4 and IL-17 were quantified using V-Plex Proinflammatory Panel 1 (human) Kit (cn: K15049D, Meso Scale Discovery), according to the manufacturer's instruction.

**Computational analysis on TCGA data.** Gene expression from acute myeloid leukaemia patients from TCGA (The Cancer Genome Atlas)[52] were obtained through the release 18 of ICGC (the International Cancer Genome Consortium, https://icgc.org/ data set name "LAML-US"). The downloaded raw read counts (RNASeqV2_RSEM_genes data) was normalized to log$_2$-counts per million with help of the functions calcNormFactors from the R package edgeR version 3.12.0[53] and voom from the R package limma version 3.26.3[54].

**Statistical analyses.** Statistical analysis was performed using *t* tests or ANOVAs or the equivalent non-parametric (Mann-Whitney or Kruskal-Wallis tests) tests when data were not normally distributed (as assessed with Kolmogorov-Smirnov test of normality and Levene's test for homogeneity of variance). Dunnett correction was eventually used for multiple comparisons. Data are shown by plotting individual data points and the mean ± s.e.m. A *p* value less than 0.05 (two-tailed) was considered as statistically significant and labelled with *. *p* values less than 0.01, 0.001 or 0.0001 were labelled respectively with **, *** or ****. Sample size was estimated with G*power software in order to obtain relevant statistical analysis ($p < 0.05$) with a power ≥ 0.8. For human experiments, the effect size was derived based on our previously published cohort of AML patients[22] focusing on ILC3 NCR$^+$ frequencies in patients vs. HD ($d = 1.79$). Thus, we estimated a minimum sample size of 10 patients at diagnosis and of 4 patients in remission ($d = 3.22$).

For mice experiments, the effect size was estimated based on a pilot experiment comparing ILC2 increase between PML-RARA APL mice and CTR mice ($d = 3.06$). Thus, we predicted that each group had to contain at least four mice per group. Mice developing unrelated diseases were excluded from the analysis. Statistical analysis of survival curves was performed with Log-rank (Mantel-Cox) test.

**Data availability**. The authors declare that all data supporting the findings of this study are available within the article and its Supplementary Information files or from the corresponding authors on reasonable request.

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

# ARTICLE

25. Kutchera, W. et al. Prostaglandin H synthase 2 is expressed abnormally in human colon cancer: evidence for a transcriptional effect. *Proc. Natl Acad. Sci. USA* **93**, 4816–4820 (1996).

26. Brown, D. et al. A PMLRARalpha transgene initiates murine acute promyelocytic leukemia. *Proc. Natl Acad. Sci. USA* **94**, 2551–2556 (1997).

27. Ablain, J. et al. Activation of a promyelocytic leukemia-tumor protein 53 axis underlies acute promyelocytic leukemia cure. *Nat. Med.* **20**, 167–174 (2014).

28. Spencer, S. P. et al. Adaptation of innate lymphoid cells to a micronutrient deficiency promotes type 2 barrier immunity. *Science* **343**, 432–437 (2014).

29. Oliphant, C. J. et al. MHCII-mediated dialog between group 2 innate lymphoid cells and CD4(+) T cells potentiates type 2 immunity and promotes parasitic helminth expulsion. *Immunity* **41**, 283–295 (2014).

30. Greenberg, N. M. et al. Prostate cancer in a transgenic mouse. *Proc. Natl Acad. Sci. USA* **92**, 3439–3443 (1995).

31. Molina, M. A., Sitja-Arnau, M., Lemoine, M. G., Frazier, M. L. & Sinicrope, F. A. Increased cyclooxygenase-2 expression in human pancreatic carcinomas and cell lines: growth inhibition by nonsteroidal anti-inflammatory drugs. *Cancer Res.* **59**, 4356–4362 (1999).

32. Hida, T. et al. Increased expression of cyclooxygenase 2 occurs frequently in human lung cancers, specifically in adenocarcinomas. *Cancer Res.* **58**, 3761–3764 (1998).

33. Hwang, D., Scollard, D., Byrne, J. & Levine, E. Expression of cyclooxygenase-1 and cyclooxygenase-2 in human breast cancer. *J. Natl Cancer Inst.* **90**, 455–460 (1998).

34. Celis, J. E. et al. Loss of adipocyte-type fatty acid binding protein and other protein biomarkers is associated with progression of human bladder transitional cell carcinomas. *Cancer Res.* **56**, 4782–4790 (1996).

35. Duffin, R. et al. Prostaglandin E(2) constrains systemic inflammation through an innate lymphoid cell-IL-22 axis. *Science* **351**, 1333–1338 (2016).

36. Carrega, P. et al. NCR(+)ILC3 concentrate in human lung cancer and associate with intratumoral lymphoid structures. *Nat. Commun.* **6**, 8280 (2015).

37. Gabitass, R. F., Annels, N. E., Stocken, D. D., Pandha, H. A. & Middleton, G. W. Elevated myeloid-derived suppressor cells in pancreatic, esophageal and gastric cancer are an independent prognostic factor and are associated with significant elevation of the Th2 cytokine interleukin-13. *Cancer Immunol. Immunother.* **60**, 1419–1430 (2011).

38. Halim, T. Y. et al. Group 2 innate lymphoid cells license dendritic cells to potentiate memory TH2 cell responses. *Nat. Immunol.* **17**, 57–64 (2016).

39. Highfill, S. L. et al. Bone marrow myeloid-derived suppressor cells (MDSCs) inhibit graft-versus-host disease (GVHD) via an arginase-1-dependent mechanism that is up-regulated by interleukin-13. *Blood* **116**, 5738–5747 (2010).

40. Di Lullo, G. et al. Th22 cells increase in poor prognosis multiple myeloma and promote tumor cell growth and survival. *Oncoimmunology* **4**, e1005460 (2015).

41. Hoechst, B. et al. A new population of myeloid-derived suppressor cells in hepatocellular carcinoma patients induces CD4(+)CD25(+)Foxp3(+) T cells. *Gastroenterology* **135**, 234–243 (2008).

42. Lallemand-Breitenbach, V. et al. Retinoic acid and arsenic synergize to eradicate leukemic cells in a mouse model of acute promyelocytic leukemia. *J. Exp. Med.* **189**, 1043–1052 (1999).

43. Ailles, L. E., Gerhard, B., Kawagoe, H. & Hogge, D. E. Growth characteristics of acute myelogenous leukemia progenitors that initiate malignant hematopoiesis in nonobese diabetic/severe combined immunodeficient mice. *Blood* **94**, 1761–1772 (1999).

44. Topalian, S. L., Drake, C. G. & Pardoll, D. M. Immune checkpoint blockade: a common denominator approach to cancer therapy. *Cancer Cell* **27**, 450–461 (2015).

45. Mager, L. F. et al. IL-33 signaling contributes to the pathogenesis of myeloproliferative neoplasms. *J. Clin. Invest.* **125**, 2579–2591 (2015).

46. Nasr, R. et al. Eradication of acute promyelocytic leukemia-initiating cells through PML-RARA degradation. *Nat. Med.* **14**, 1333–1342 (2008).

47. Hasan, M. et al. Semi-automated and standardized cytometric procedures for multi-panel and multi-parametric whole blood immunophenotyping. *Clin. Immunol.* **157**, 261–276 (2015).

48. Rabinovich, B. A. et al. Visualizing fewer than 10 mouse T cells with an enhanced firefly luciferase in immunocompetent mouse models of cancer. *Proc. Natl Acad. Sci. USA* **105**, 14342–14346 (2008).

49. Soudja, S. M. et al. Tumor-initiated inflammation overrides protective adaptive immunity in an induced melanoma model in mice. *Cancer Res.* **70**, 3515–3525 (2010).

50. Bronte, V. et al. Recommendations for myeloid-derived suppressor cell nomenclature and characterization standards. *Nat. Commun.* **7**, 12150 (2016).

51. Gupta, B. et al. Simultaneous coexpression of memory-related and effector-related genes by individual human CD8 T cells depends on antigen specificity and differentiation. *J. Immunother.* **35**, 488–501 (2012).

52. Cancer Genome Atlas Research, N.. Genomic and epigenomic landscapes of adult de novo acute myeloid leukemia. *N. Engl. J. Med.* **368**, 2059–2074 (2013).

53. Robinson, M. D., McCarthy, D. J. & Smyth, G. K. edgeR: a Bioconductor package for differential expression analysis of digital gene expression data. *Bioinformatics* **26**, 139–140 (2010).

54. Ritchie, M. E. et al. limma powers differential expression analyses for RNA-sequencing and microarray studies. *Nucleic Acids Res.* **43**, e47 (2015).

## Acknowledgements

We are grateful to the patients for their dedicated collaboration and to healthy donors for their blood or CD34+ HSC donation. We thank Prof. P. Romero for insightful discussions and support of this study. We thank Prof. H. de Thé for providing the PML-RARA APL mice, Dr A. Donda for help with the TRAMP model, P. Reichenbach and S. Ferreira Lopes for technical help. We thank the caretakers of the animal facility for their excellent assistance. We thank Daniela Pende (Istituto di Ricovero e Cura a Carattere Scientifico Azienda Ospedaliera Universitaria San Martino-Istituto Nazionale per la Ricerca sul Cancro, Genoa, Italy), Alessandro Moretta (Dipartimento di Medicina Sperimentale, Università degli Studi di Genova, Genoa, Italy) and Silvia Parolini (Sezione di Oncologia e Immunologia Sperimentale, Dipartimento di Medicina Molecolare e Traslazionale, Università di Brescia) for providing masking antibodies against NKp30 (clone F252) and NKp46 (clone KL247). This work was supported in part by grants from the Swiss National Science Foundation (Ambizione PZOOP3_161459), Fondazione San Salvatore, ProFemmes UNIL, Fondation Pierre Mercier pour la Science, the Swiss Cancer League KFS-3710-08-2015-R to C.J.; by the Swiss National Science Foundation (Marie Heim Vögtlin fellowship PMPDP3_164447) to S.T.; by the Ludwig Institute for Cancer Research (LICR) to G.C. and D.V.; by the Swiss National Foundation 32003B_146638, the Novartis Foundation for medical-biological Research 15C165 and the Foundation for the Fight against Cancer #369 and #324 to L.D.

## Author contributions

S.T., M.F.C., A.M.-U., B.S., A.G.-C., M.L., V.S., G.V., I.P., C.G., P.B., M.G., R.P., P.F. and C.J. performed the experiments; C.P., H.M., E.B., E.O.L., G.M.B., C.C.-S., D.T., A.S., O.S., A.R., E.G., G.B., C.T., F.C., C.A.A., L.M., S.M., P.G.P., E.M., A.N.J.M., D.V., G.C., D.M., A.C. and L.D. provided reagents and patients' samples; J.R. and D.G. conducted the analysis of TCGA data; S.T., M.F.C., A.M.-U., B.S., A.G.C., D.V., G.C., L.D. and C.J. designed research, analysed the experiments, discussed the results and wrote the manuscript. A.M.-U. and A. G.-C. contributed equally to this work.

## Additional information

**Competing interests:** The authors declare no competing financial interests.

**13**

Sara Trabanelli[1], Mathieu F. Chevalier [2], Amaia Martinez-Usatorre[1], Alejandra Gomez-Cadena[1], Bérengère Salomé[1], Mariangela Lecciso[3], Valentina Salvestrini[3], Grégory Verdeil[1], Julien Racle [1,4], Cristina Papayannidis[3], Hideaki Morita[5,6], Irene Pizzitola[1], Camille Grandclément[1], Perrine Bohner[2], Elena Bruni[7,8], Mukul Girotra [1], Rani Pallavi[9], Paolo Falvo[9], Elisabeth Oppliger Leibundgut[10], Gabriela M. Baerlocher[10], Carmelo Carlo-Stella[11,12], Daniela Taurino[11,12], Armando Santoro[11,12], Orietta Spinelli[13], Alessandro Rambaldi[13,14], Emanuela Giarin[15], Giuseppe Basso[15], Cristina Tresoldi[16], Fabio Ciceri[17], David Gfeller[1,4], Cezmi A. Akdis[5], Luca Mazzarella[9,18], Saverio Minucci[9], Pier Giuseppe Pelicci[9], Emanuela Marcenaro [19], Andrew N.J. McKenzie[20], Dominique Vanhecke[1], George Coukos[1], Domenico Mavilio [7,8], Antonio Curti[3], Laurent Derré [2] & Camilla Jandus [1]

[1]Ludwig Institute for Cancer Research, Department of Fundamental Oncology, University of Lausanne, Biopole 3-02DB61, Ch. Des Boveresses 155, CH-1066 Epalinges, Switzerland. [2]Urology Research Unit, Lausanne University Hospital (CHUV), 1011 Lausanne, Switzerland. [3]Department of Specialistic, Diagnostic and Experimental Medicine, Institute of Hematology "Seràgnoli", University of Bologna, 40138 Bologna, Italy. [4]Swiss Institute of Bioinformatics (SIB), 1015 Lausanne, Switzerland. [5]Swiss Institute of Allergy and Asthma Research (SIAF), University of Zurich, 7270 Davos, Switzerland. [6]Christine Kühne-Center for Allergy Research and Education, 7265 Davos, Switzerland. [7]Department of Medical Biotechnologies and Translational Medicine, University of Milan, 20133 Milan, Italy. [8]Unit of Clinical and Experimental Immunology, Humanitas Clinical and Research Center, 20089 Rozzano-Milan, Italy. [9]Department of Experimental Oncology, European Institute of Oncology, 20139 Milan, Italy. [10]Department of Hematology, Bern University Hospital, University of Bern, 3010 Bern, Switzerland. [11]Humanitas Cancer Center, Humanitas Clinical and Research Center, 20089 Rozzano-Milan, Italy. [12]Department of Biomedical Sciences, Humanitas University, 20089 Rozzano-Milan, Italy. [13]Hematology and Bone Marrow Transplant Unit, Ospedale Papa Giovanni XXIII, 24127 Bergamo, Italy. [14]Università Statale di Milano, 20122 Milan, Italy. [15]Dipartimento per la Salute della Donna e del Bambino, Clinica di Oncoematologia Pediatrica, University of Padova, 35128 Padova, Italy. [16]Immunoematologia e Medicina Trasfusionale, Laboratorio Ematologia Molecolare, Biobanca Neoplasie Ematologiche, San Raffaele Hospital, 20132 Milano, Italy. [17]Divisione di Ricerca di Medicina Rigenerativa, Terapia Cellulare e Genica IRCCS, San Raffaele Hospital, 20132 Milano, Italy. [18]Division of Innovative Therapies, European Institute of Oncology, 20141 Milan, Italy. [19]Department of Experimental Medicine (DI.ME.S.)-Section of Histology, and Center of Excellent of Biomedical Research (CEBR), University of Genoa, 16132 Genoa, Italy. [20]MRC Laboratory of Molecular Biology, Cambridge CB2 0QH, UK. Sara Trabanelli and Mathieu F. Chevalier contributed equally to this work. Laurent Derré and Camilla Jandus jointly supervised this work.

