## [Peer Review File · Nature Communications]

Reviewer #1 (Remarks to the Author):

This manuscript describes a novel tumor immunosuppressive axis consisting of ILC2 and M-MDSC. They provide evidence that this axis functions in APL by showing in various models that targeting PGD2, IL-13 and NKp30 results in increased survival of mice exposed to APL like cells.

This is a very interesting manuscript. The authors have adequately addressed the comments of the reviewers of the first version. The new experiments with the antiIL-13 and anti NKp30 antibodies present a significant improvement.

One comment; the authors mention in their comment to reviewer 3 that they don't claim that ATRA acts directly on the ILC2-M-MDSC axis. However Belkaid and colleagues have demonstrated that RA acts directly on ILC2 reducing the numbers of these cells in the gut while increasing ILC3 (Spencer et al Science 24 Jan 2014:Vol. 343, Issue 6169, pp. 432-437) The authors may include this in their discussion.

They may consider to directly test the effects of ATRA on human ILC2 activation and cytokine production in vitro

Reviewer #2 (Remarks to the Author):

The transferred manuscript "Tumor-derived PGD2 and NKp30-B7H6 engagement drives an immunosuppressive ILC2-MDSC axis' by Trabanelli et al." with appropriate modification presents a new idea of how ILC2-MDSC axis may involve and potentially exploited for cancer therapeutics. While mouse models have been used to study and demonstrate its therapeutic potentials, they have rather limited relevance to some of the major hypotheses and the human disease due to its lack of NKp30 expression. Therefore, one would like to highlight the importance of testing the relevant hypothesis in primary APL cells or at least human APL cell line expressing PML-RARa fusion in an in vivo setting in the future study.

RE: PBP reply NCOMMS-17-11034-T

Reviewers' Comments:

Reviewer #1:

None

Answer: We thank the reviewer for his/her appreciation of our extensively revised manuscript.

Reviewer #3:

In the revised manuscript 'Tumor-derived PGD2 and NKp30-B7H6 engagement drives an immunosuppressive ILC2-MDSC axis' by Trabanelli et al., the authors have provided a large volume of new data to address various concerns of the reviewers. While these new data have clearly improved various aspects of manuscript, the in vivo data key to the role of ILC2-MDSC in APL/prostate cancer pathogenesis are still under-developed and inconclusive. The reviewer is still not convinced that targeting ILC2-MDSC is a valid therapeutic approach for APL or prostate cancer.

Specific comments:

While the authors have now provided new in vivo data in Figure 5, they are rather preliminary and fail to support the main conclusion. Firstly, in the iCOS-T model, all has been shown in Figure 5a-c was a reduction of M-MDSC upon removal of ILC2. It does not have any data on APL pathogenesis and treatment response. They should show the impact on survival and most importantly if the phenotype can be rescued by activating or putting back ILC2/M-MDSC.

Answer: We thank the reviewer for raising this point. The use of iCOS-T mice for transient, partial depletion of ILC2 by administration of diphtheria toxin (Oliphant CJ et al., Immunity, 2014) was specifically used to answer to previous points raised by the reviewers regarding the reciprocal interactions between ILC2 and M-MDSC. Therefore, we performed this experiment not to assess survival, but to verify if in animals with reduced levels of ILC2 we would observe a reduction in M-MDSC. This is indeed the case, arguing for a direct link between these two cell types, as shown in main **Figure 5a-c**.

To test the role of ILC2 and M-MDSCs on tumor progression and survival, we have used settings that we considered therapeutically more relevant, e.g. the APL mouse model using PGD2 inhibitor and blocking anti-IL-13 antibodies, and the humanized HIS APL model using PGD2 inhibitor, blocking anti-IL-13 and anti-NKp30 antibodies. In both these models, we observed normalization of ILC2 and M-MDSCs, accompanied by survival advantage in the treated compared to control mice (**Figure 5d-k**).

This is a very incomplete study and where are the rest of the data. Without them, it is difficult to interpret the result and draw a conclusion.

Answer: we apologize with the reviewer for the misperception related to the presentation of our *in vivo* data. In fact, all originally obtained data, rather than a selection, are shown in the manuscript Figures. Every one of the new experiments in the revised manuscript was designed to address the questions from the reviewers. To clarify this point, we have revised the main text of the manuscript to better describe which was the purpose of the different *in vivo* experiments and rephrased the conclusions throughout the manuscript.

Then the authors subjected a FVB APL model to PGD2 inhibitor and anti-IL13 treatments (Figure 5d-h). In spite of significant reduction of ILC2 and M-MDSC under these treatments, the survival of treated mice hardly showed any meaningful growth advantage. They all died within 16 days in regarding the treatments. In contrast to their main conclusion, it suggests minimal if any therapeutic value of suppressing ILC2/M-MDSC for APL treatment.

Answer: the FVB APL mouse model is the first model, in which we have tested the therapeutic relevance of our findings in the originally submitted manuscript. Given that the NKp30 receptor is not expressed in mice, we used this model to therapeutically interfere with the action of PGD2 and IL-13 by using a PGD2 inhibitor and blocking anti-IL-13 antibodies. In this model, the *in vivo* partial normalization of PGD2, ILC2 and M-MDSC was accompanied by a significant delay in APL progression. However, based on the facts that (i) this model is very aggressive and that (ii) our *in vitro* experiments showed a key role of the B7H6-NKp30 interaction for human ILC2 activity, the observed survival advantage was only modest. Therefore, we moved to the more relevant model of mice reconstituted with a human immune system (HIS).

Finally, the authors switched to yet another APL model, HL60 cell line with least pathological relevance to human APL, as it does not even express the RARa fusion, a defining feature for APL. By applying a combination of PGD2 inhibitor, anti-IL13 and anti-NKp30 antibodies to this model, the authors can now detect a significant survival benefit (Figure 5i-n).

Answer: we agree with the reviewer that with the HIS mice we are providing yet an additional mouse model. However, we believe that this model has a crucial relevance to test directly *in vivo* the therapeutic potential of targeting the ILC2-M-MDSC-IL-13 axis. Indeed, HIS mice are mice reconstituted with a human immune system, thus representing the only model allowing us to test the *in vivo* relevance of the B7H6-NKp30 interaction in the identified immunosuppressive axis (NKp30 not being expressed in mice).

Regarding the use of the HL60 cell line, while we do agree that this line does not express the PML-RAR fusion protein, it is known to possess typical characteristics of APL and to be responsive to ATRA (https://www.lgcstandards-atcc.org/Products/All/CCL-240.aspx?geo_country=ch). Moreover, we showed in our *in vitro* studies that the HL-60 cell line had comparable functional features (PGD2 secretion; B7H6 expression, **Figure 2**) as the NB4 cell line, an established human PML-RAR+ APL cell line. Based on the availability of an HL-60 cell line transfected with enhanced luciferase, thus allowing careful non-invasive direct monitoring of leukemia progression and treatment effectiveness, we used this line for *in vivo* experiments as an optimal surrogate of APL (such as NB4 cells).

By applying a combination of PGD2 inhibitor, anti-IL13 and anti-NKp30 antibodies to this model, the authors can now detect a significant survival benefit (Figure 5i-n). However, if we examined the effect of these combo treatments on level of ILC2 and M-MDSC, they resulted in a very similar reduction as observed with PGD2 inhibitor plus anti-IL13 treatments in Figure 5f-g. The latter however has minimal effect on survival. These results will again go against

their main conclusion, and instead suggest that the role of ILC2 and M-MDSC on treatment response is rather minimal.

Answer: even if we accept that comparison of treatment efficacy between different tumor mouse models is a useful way to evaluate treatment responses in tumor immunology, in this case the reviewer is comparing data originating from a murine tumor model (**Figure 5d-h**) with those obtained in humanized mice bearing tumors of human origin (**Figure 5i-n**). We are afraid we have to respectfully disagree on drawing conclusions from these comparisons. The *in vivo* tumor cell growth kinetics are sharply different, as are the tumor cell biology and ILC biology (no NKp30 in mice) in these different models.

The NKp30 blocking antibody can well be targeting other key cell types, which has not been considered/excluded by authors.

Answer: we thank the reviewer for raising this point. Indeed, as previously shown by us and others, triggering of NKp30 in NK cells by the engagement of B7H6 or other unknown ligands on target cells (e.g. tumor cells; dendritic cells) results in human NK cell activation. The masking, anti-human NKp30 antibody used in our study was extensively shown to reduce human NK cell activity (Pesce S. et al., *Oncoimmunology*, 2015; Brandt CS., et al., *JEM*, 2009; Ferlazzo G., et al, *JEM*, 2002). Thus, we monitored the frequency of NK cells in HIS mice, but as previously reported by others, NK cell reconstitution in these animals is very poor and reconstituted NK cells would require pre-activation to acquire functional competence (Strowig T., *Blood*, 2010). Therefore, the fact that in our *in vivo* HIS APL model we observe a restoration of tumor immunity upon treatment by blocking anti-NKp30 antibodies argues for an action of this antibody on NKp30 expressed by ILC2, resulting in reduced *in vitro* secretion of IL-13, as previously reported (Salimi M., et al., *Ji*, 2016), rather than inhibiting the activity of NK cells. This would indeed result in reduced leukemic control.

On the other hand, the *in vivo* data on prostate cancer is even less convincing, none of the equivalent work done in APL (**Figure 5**) was carried out in the prostate cancer study. Therefore, the reviewer is not convinced by the presented data that ILC2 and M-MDSC play an active role in APL pathogenesis and could be therapeutic targets. Again, the evidence for prostate cancer is even less convincing.

Answer: we agree with the reviewer that the provided amount of *in vivo* data for prostate cancer is reduced compared to APL data. However, to the best of our knowledge, there are no established models of spontaneous human prostate cancer in HIS mice, suitable to test the *in vivo* therapeutic relevance of our *ex-vivo* and *in vitro* observations. Given the need to confirm in the future our observations in novel *in vivo* models, we have toned down throughout the manuscript our conclusions that proposed the targeting of this novel immunosuppressive axis as a novel intervention in different malignancies.

Reviewer #4:

Remarks to the Author:

This manuscript describes a novel tumor immunosuppressive axis consisting of ILC2 and M-MDSC. They provide evidence that this axis functions in APL by showing in various models

that targeting PGD2, IL-13 and NKp30 results in increased survival of mice exposed to APL like cells.

This is a very interesting manuscript. The authors have adequately addressed the comments of the reviewers of the first version. The new experiments with the antiIL-13 and anti NKp30 antibodies present a significant improvement.

Answer: we thank the reviewer for acknowledging our efforts of revising the original manuscript.

One comment; the authors mention in their comment to reviewer 3 that they don't claim that ATRA acts directly on the ILC2-M-MDSC axis. However Belkaid and colleagues have demonstrated that RA acts directly on ILC2 reducing the numbers of these cells in the gut while increasing ILC3 (Spencer et al Science 24 Jan 2014:Vol. 343, Issue 6169, pp. 432-437) The authors may include this in their discussion.

Answer: we thank the reviewer for raising this point. By evaluating the *in vivo* effect of ATRA in HIS mice, we did not observe an effect on ILCs in the circulation or in the bone marrow. However, as mentioned by the reviewer, others reported on a direct effect of ATRA on ILC in the gut. As asked by the reviewer, we have added the relevant reference by Spencer et al in the main manuscript text (page 9).

They may consider to directly test the effects of ATRA on human ILC2 activation and cytokine production *in vitro*.

Answer: as asked by the reviewer we have exposed *in vitro* 2 highly-pure human ILC2 short-term expanded lines to activating stimuli (IL2, IL-33, PGD2) alone or in combination with ATRA, and monitored their phenotype as well as IL-13 secretion by flow cytometry. As shown in the Figure here below, we did not observe conversion of ILC2 to ILC3 (as assessed by CRTH2⁻ckit⁺), and we did not observe significant differences in IL-13 secretion by the addition of ATRA.

Figure Legend. Highly-pure human ILC2 short-term expanded lines (Lin⁻CD127⁺CRTH2⁺) have been treated overnight by IL-2 (50 U/ml), IL-33 (50 ng/ml), PGD2 (100 nM) and ATRA (100 nM), as indicated in the graph legend. 2 µg/ml of BrefeldinA was added to the cultures. Conversion of ILC2 into ILC3 was monitored by evaluating the frequency of CRTH2⁻ckit⁺ cells (left graph side), and IL-13 production was assessed (right graph side) by multicolour flow cytometry.